# Magic number colloidal clusters as minimum free energy structures

Junwei Wang[1], Chrameh Fru Mbah[2], Thomas Przybilla[3], Benjamin Apeleo Zubiri[3], Erdmann Spiecker[3], Michael Engel [2] & Nicolas Vogel [1]

Clusters in systems as diverse as metal atoms, virus proteins, noble gases, and nucleons have properties that depend sensitively on the number of constituent particles. Certain numbers are termed 'magic' because they grant the system with closed shells and exceptional stability. To this point, magic number clusters have been exclusively found with attractive interactions as present between atoms. Here we show that magic number clusters exist in a confined soft matter system with negligible interactions. Colloidal particles in an emulsion droplet spontaneously organize into a series of clusters with precisely defined shell structures. Crucially, free energy calculations demonstrate that colloidal clusters with magic numbers possess higher thermodynamic stability than those off magic numbers. A complex kinetic pathway is responsible for the efficiency of this system in finding its minimum free energy configuration. Targeting similar magic number states is a strategy towards unique configurations in finite self-organizing systems across the scales.

[1] Institute of Particle Technology, Friedrich-Alexander University Erlangen-Nürnberg, 91058 Erlangen, Germany. [2] Institute for Multiscale Simulation, Friedrich-Alexander University Erlangen-Nürnberg, 91058 Erlangen, Germany. [3] Institute of Micro- and Nanostructure Research, Friedrich-Alexander University Erlangen-Nürnberg, 91058 Erlangen, Germany. Correspondence and requests for materials should be addressed to M.E. (email: michael.engel@fau.de) or to N.V. (email: nicolas.vogel@fau.de)

The structure of particle clusters is strongly affected by geometric constraints. This first became apparent in the study of atomic nuclei. Nucleons preferentially arrange into shells, which lead to the identification of the series of numbers that permit closed shells as magic numbers[1,2]. Deviations from a pure statistical distribution of cluster sizes are also observed in the mass spectrum of small noble gas and metal clusters[3,4]. Clusters with atom numbers that follow magic numbers show enhanced stability as a result of the maximized numbers of neighbors[5–7]. This tendency provides a driving force to form clusters with closed, concentric shells, including the well-known Mackay icosahedra[8]. Atomic clusters therefore often possess complex geometries that differ from the typical face-centered cubic (FCC) bulk crystal structure[9,10].

Whereas the appearance and structure of atomic clusters are commonly explained by potential energy minimization, especially at low temperature[11], the formation of colloidal clusters is typically governed by several factors. Colloids are initially stabilized against aggregation, which prevents them from aggregating in solution[12,13]. However, colloids can self-assemble into clusters by increasing short-range attraction[14–17] or by applying geometric confinements[18–26]. Phenomena that contribute to the development of colloidal clusters during a confined self-assembly process include the interaction of soft ligand shells[27], the presence of depletants[17], capillary forces acting on particles during the drying process[19], and entropy maximization[28]. The latter case is comparably easy to control in the experiment because it only requires weakly interacting colloids, which are described well via the simple hard sphere model[29,30]. Temperature as the control variable for atomic cluster formation is then replaced by the packing fraction of the hard spheres[31–35].

Small numbers of nanoparticles and colloids pack into defined clusters with a wide variety of symmetries[17,36–38]. With increasing particle number, entropy maximization has been identified as the driving force favoring the formation of icosahedral symmetry of spherical colloidal clusters[28,39]. In such confined systems, the interface boundary acts as a source for heterogeneous nucleation[40] of crystalline patches[41] that subsequently expands towards the cluster center[28]. If the number of particles within the confining element is further increased, the effect of confinement is continuously reduced until, from a system size of approximately 100,000 particles, bulk crystal structures with FCC symmetry are retained[28]. Despite these general insights, the detailed structure of a large colloidal cluster formed from a discrete number of colloidal particles remains unknown. While the dominant occurrence of atomic clusters with magic numbers in mass spectra[3,4] suggests self-adjustment in their sizes during formation by addition or removal of atoms to reach magic numbers under appropriate circumstances, the number of particles in a colloidal cluster remains fixed in confinement, which allows experiments to explore clusters of arbitrary sizes. This raises the fundamental question of whether some colloidal cluster sizes are thermodynamically preferred over others.

Here, we demonstrate that the magic number phenomenon known from atomic clusters extends to the colloidal realm. We report a discrete family of icosahedral clusters spanning a large range of particle numbers and propose a geometric model based on the Mackay and anti-Mackay clusters to accurately describe all observed cluster structures. Simulations and high-precision free energy calculations reproduce and explain the experimental observation.

## Results

**Fabrication of colloidal clusters**. We study the system size-dependent colloidal cluster formation in monodispersed droplets of an aqueous dispersion of polystyrene (PS) colloidal particles with different concentrations in a continuous oil phase produced by microfluidics (Supplementary Figure 1). The PS colloidal particles are 244 nm in diameter and stabilized by carboxylate surface groups introduced as comonomers in the synthesis by surfactant-free emulsion polymerization. Over the course of water evaporation, the volume fraction of the colloidal particles gradually increases towards a solidified colloidal cluster. Four cluster morphologies prevail (Fig. 1a–d). With the fastest evaporation, buckled clusters form as the droplet interface moves faster than colloidal particles can consolidate (Fig. 1a, Supplementary Figure 2). When the evaporation rate is lowered, spherical clusters dominate (Fig. 1b). Spherical clusters exhibit a uniformly curved surface with only weak crystalline order and are the morphology most frequently reported in the literature[18,42–45]. Partial icosahedral clusters with incompletely developed five-fold symmetry axes at the surface (Fig. 1c) form with further decreasing evaporation rate. Very slow evaporation provides sufficient time for the colloidal particles to arrange into icosahedral clusters (Fig. 1d), which are characterized by a fully developed pattern of five-fold axes at the surface. The uniformity of the colloidal clusters (Fig. 1e, f) enables us to statistically evaluate the evaporation rate-dependent cluster formation: the fraction of icosahedral clusters increases up to 75% as evaporation is slowed down (Fig. 1g). Similarly, at the lowest evaporation rate, the dominant species of the observed clusters evolve from buckled to spherical to icosahedral symmetry with increasing assembly time (Fig. 1h, Supplementary Figure 3).

**Geometric model of magic number colloidal clusters**. Well-formed icosahedral clusters are characterized by distinct surface features. The surface is tiled with rectangles and truncated triangles, which alternate around five-fold axes. Together, these structural elements form a closed shell spanning the cluster's surface (Fig. 2a). In analogy to atomic clusters with complete outer shells[3,4], we term such clusters as magic number colloidal clusters (MCCs). To rationalize the appearance of MCCs, we recall that 12 identical spheres readily arrange into an icosahedron around a central sphere. Subsequent Mackay shells can be added concentrically[8]. Figure 2b exemplarily shows a 10-shell Mackay icosahedron as the starting point for a model to describe the MCC structure. It consists of 20 slightly deformed tetrahedra with FCC structure sharing a central sphere, each twinned with three neighboring tetrahedra. We expand the Mackay core by adding anti-Mackay shells[46–50] that consist of 20 additional tetrahedra over the icosahedron faces, filling their gaps with 30 tetrahedra over the icosahedron edges, and finishing the construction with 60 tetrahedra over the icosahedron vertices. The resulting Pentakis dodecahedron model[51] maintains icosahedral symmetry and consists of 130 twinned tetrahedra. Finally, we apply spherical truncation to remove spheres in the model whose distances to the center is larger than the truncation radius. This mimics the effect of confinement of the confined colloids, which are forced into a spherical shape by the geometry of the emulsion droplet.

Although complex at first glance, MCCs are determined by only two parameters: the number of shells of the Mackay core $m$ and the number of anti-Mackay shells $a$, which is controlled by the truncation radius (Fig. 2c). We introduce a notation to classify the MCCs as $(m + a)_a$ types, specifying the total number of shells, $m + a$, and the number of anti-Mackay shells $a$. The distinct surface patterns of the MCCs are directly correlated with their crystal structure and can be used to deduce the number of Mackay and anti-Mackay shells. From the model, we identify that the rectangular surface features are the characteristic structural element associated with the anti-Mackay shells (Fig. 2b). They result from truncation of the 20 twinned tetrahedra added to the facets of the icosahedral structure and the additional 90

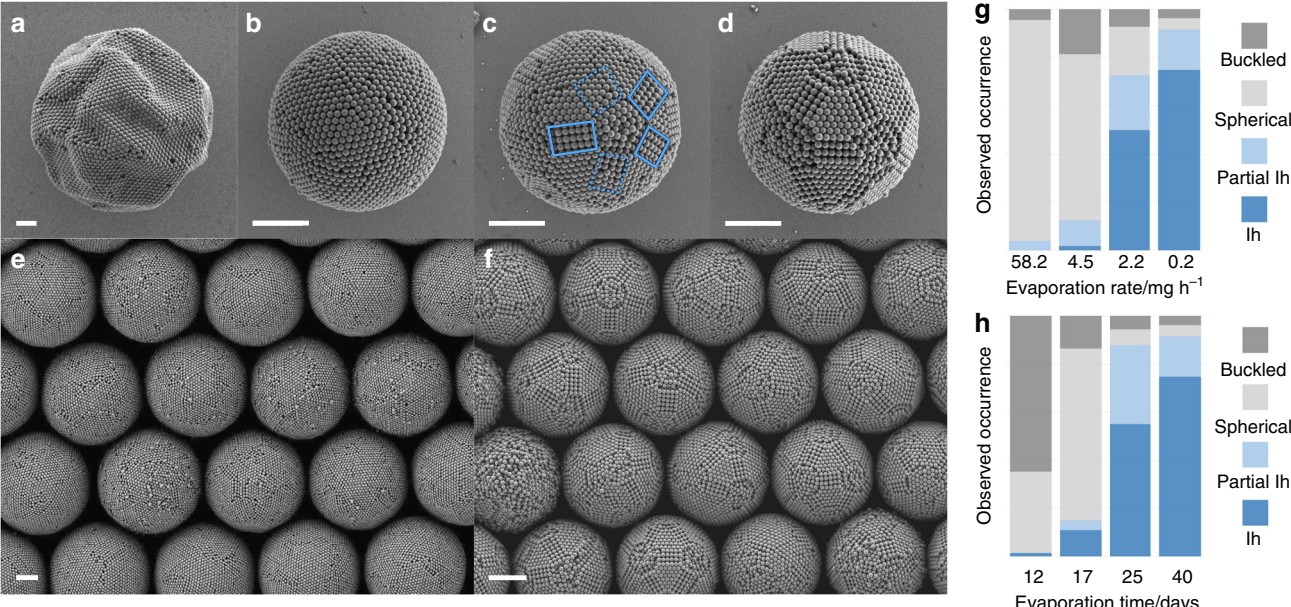

**Fig. 1** Colloidal clusters from confined self-assembly in water-in-oil emulsion droplets. Four distinct cluster morphologies with increasing degree of ordering are observed: **a** buckled clusters partially collapse upon evaporation into non-spherical shape; **b** spherical clusters exhibit only local order; **c** partial icosahedral clusters show one or more five-fold symmetry axes and incomplete faceting (dotted blue boxes); **d** icosahedral clusters have well-defined facets, edges, and vertices and complete icosahedral symmetry. **e, f** Low-magnification scanning electron microscopy (SEM) images highlight the uniformity in size and structure of the prepared clusters. Spherical and icosahedral clusters dominate in the limit of fast (**e**) and slow (**f**) evaporation, respectively. **g, h** Statistical evaluation of the observed morphologies as a function of the evaporation rate (**g**) and as an evolution over time for the slowest evaporation rate (**h**) showing the progression from spherical to icosahedral (Ih). Scale bars, 2 μm

tetrahedra filling the open voids of the model structure. In the absence of an anti-Mackay shell, a single line of colloids marks the facet of the icosahedron. A single anti-Mackay shell produces a rectangle with a width of two particles, two anti-Mackay shells a width of three particles (as in the example structure) and so on. An equation of the relationship between surface and interior shell structure is given in Supplementary Figure 4. Python code to generate MCC models following the procedure in Fig. 2b is included in Supplementary Information. By comparison with the model, we can now identify the MCC in Fig. 2a as of $12_2$ type, i.e., consisting of 10 Mackay and 2 anti-Mackay shells.

The geometry of the Pentakis dodecahedron necessitates deformation on all tetrahedral grains in the model (Fig. 2d). Grains in the Mackay core and over anti-Mackay faces are identical and least deformed, grains over anti-Mackay edges and vertices are more deformed. The deformation analysis of our model suggests particles along the width of the rectangles in the cluster surface are separated by 1.13 times their diameter, compared to those along the length of the rectangle by 1.05, which can be observed experimentally (Fig. 2a, Supplementary Figures 5, 8). It also suggests extra particles are prone to accumulate in the vertices regions where tetrahedral grains have the highest degree of deformation. Our model extends the well-established Mackay icosahedron[52] to a new sphere packing model into a Pentakis dodecahedron, which may help the understanding of binary clusters where slightly larger component resides in more deformed regions[51,53,54]. Note that the icosahedral order of the model and MCCs originates from multi-twinned tetrahedral grains with a deformed FCC crystal structure, which bears no direct connection to quasicrystals.

**Electron tomographic confirmation of the model**. We employ electron tomography of a $7_2$ type MCC to resolve its three-dimensional structure. The cluster structure and model agree quantitatively, as seen by the similarity of bright field scanning

transmission electron microscopy (STEM) images and the semi-transparent model images in projections along the symmetry axes (Fig. 3a–c, f–h). This agreement confirms that the MCC indeed is crystalline and that the cluster type deduced from the surface pattern is consistent with its internal structure. The recorded tilt series underlines the icosahedral nature of the cluster. During rotation, views along two-, three-, and five-fold symmetry axes gradually transition from one to another (Supplementary Movie 1, Supplementary Movie 2). In the tomographic reconstruction of the data, characteristic surface features (Fig. 3d, i, Supplementary Figure 5) and details of the interior (Fig. 3e, j, Supplementary Movie 3) coincide between experiment and model. In the cross-section of the reconstructed cluster, two triangles of 21 hexagonally closed-packed particles sharing a single particle at the center can be clearly distinguished (Fig. 3e, j). These particles form the triangular faces of two tetrahedra in a five-shell Mackay icosahedron. Four particles at the surface belong to the second anti-Mackay shell (indicated by a green line), which enables us to identify the cluster as a $7_2$ type (Supplementary Figure 6).

**Library of MCCs**. From the analysis of hundreds of clusters of size ranging from 100 to 10,000 colloidal particles, we generate a library of experimentally observed MCCs (Fig. 4, Supplementary Figure 7). All these clusters exhibit complete, well-defined outer shells and can be assigned a magic number cluster type of the form $(m + a)_a$, deduced from their surface features. In all cases, experimental observation and model coincide. The model predicts that the number of colloids per cluster follows the approximate relationship $N = \frac{10}{3}(m + a)^3$ derived for Mackay clusters[8]. Furthermore, the number of colloidal particles per cluster decreases from type $(m + a)_a$ to type $(m + a)_{a+1}$, i.e., when a Mackay shell converts to an anti-Mackay shell (Fig. 4j–l), as a result of lower local density in anti-Mackay shells. Figure 4m summarizes all

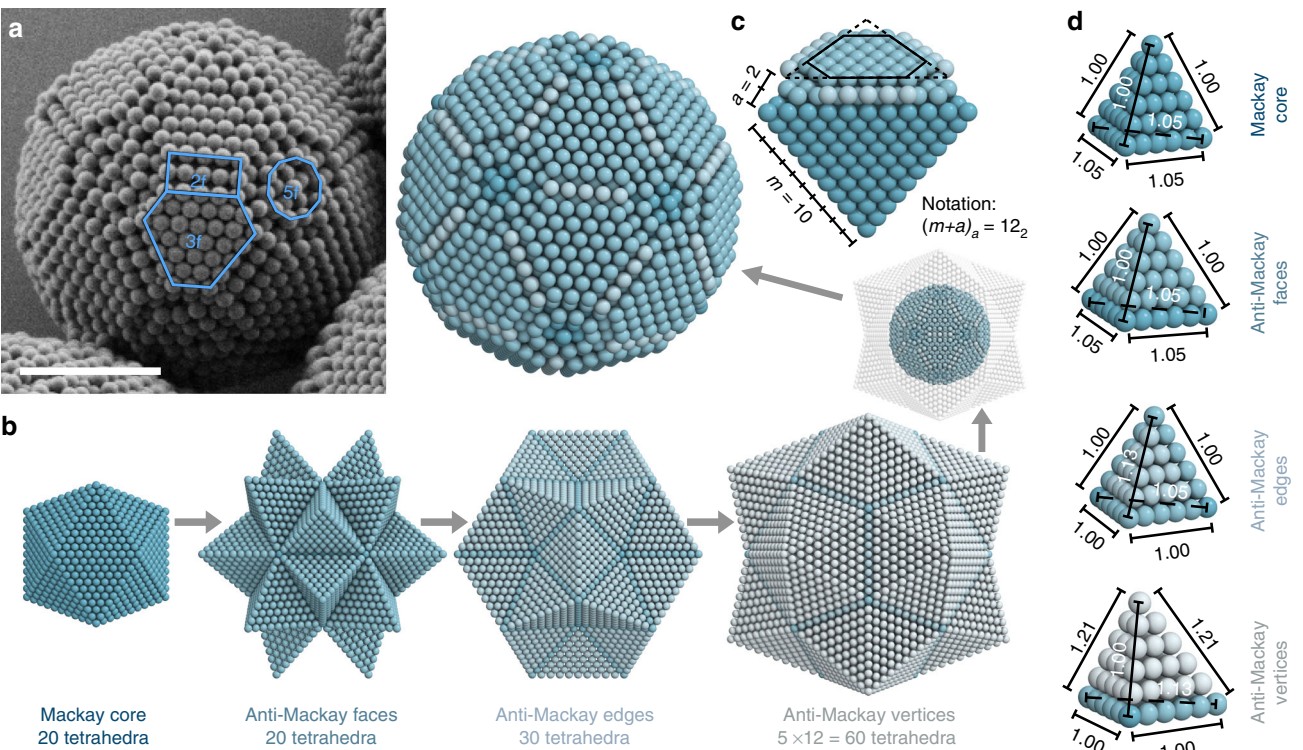

**Fig. 2** Model of magic number colloidal clusters. **a** SEM image of a magic number colloidal cluster (MCC). The surface is tiled with 5 × 3 rectangles (along two-fold symmetry axes, "2f"), truncated triangles ("3f"), and concentric rings around five-fold axes ("5f"). Scale bar, 2 μm. **b** Model of a MCC. A Mackay icosahedron core is expanded by adding twinned tetrahedral grains over the core icosahedral faces, edges, and vertices, forming anti-Mackay shells. The outer geometry of the complete model is a Pentakis dodecahedron. The effect of droplet confinement is mimicked by applying spherical truncation to remove some spheres in the model. The resulting structure accurately reproduces the experimentally observed MCC in (**a**). **c** Building block of the model with 10 Mackay and 2 anti-Mackay shells. MCCs are denoted as of $(m + a)_a$ type, where $m$ is the number of Mackay shells and $a$ the number of anti-Mackay shells. **d** Tetrahedral grains in the different parts of the model with marked scale factors compared to a perfect FCC grain. All grains deviate from regular tetrahedra as a consequence of strains in the icosahedral structure. Grains in anti-Mackay edges and vertices are deformed more strongly

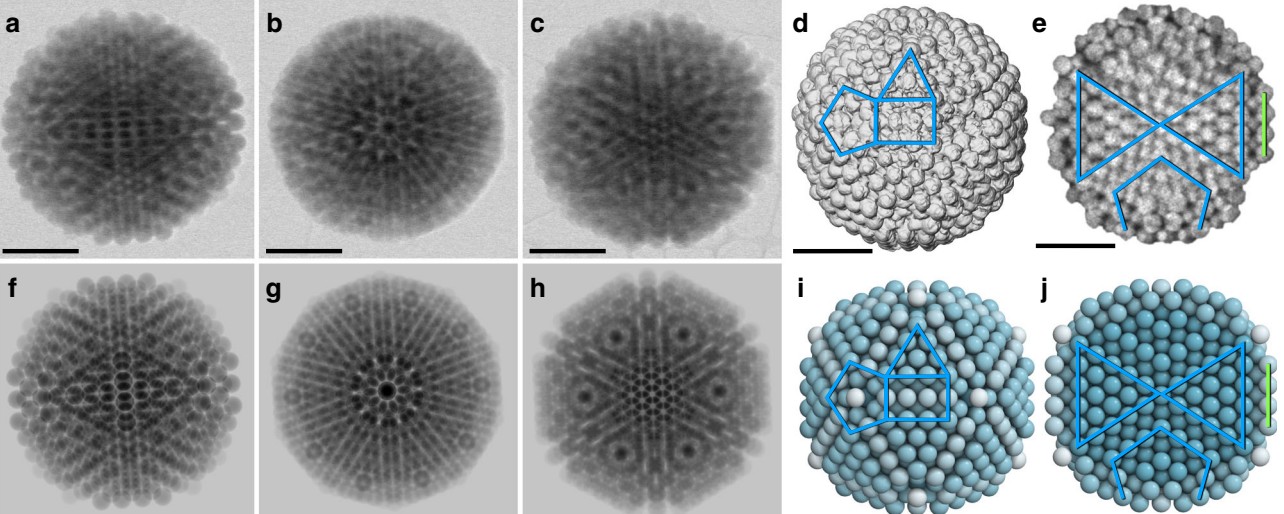

**Fig. 3** Electron microscopy tomography confirmation of model. **a–c** Scanning transmission electron microscopy (STEM) bright-field images of a magic number colloidal cluster viewing along two-fold, five-fold, and three-fold axes, characteristic for icosahedral symmetry. Two staggered 10-spike rings in the center along the five-fold axis are clearly seen in (**b**). **d** STEM tomography reconstruction provides a three-dimensional visualization of characteristic surface features of triangles with four particles at side, rectangles of three and four particles at width and length, and pentagon with three particles at side (indicated in blue). **e** Cross-sectional views through the reconstruction at the central plane reveals the characteristic internal structure. A section of the Mackay core is marked by two blue triangles. The vertex region is marked by its characteristic pentagon. The second anti-Mackay shell is indicated by a green line segment. Scale bars, 1 μm. **f–j** The experimental images show excellent agreement with $7_2$ MCC type model reconstructions using semi-transparent spheres mimicking the STEM imaging process (**f–h**), external (**i**) and cross-sectional views (**j**)

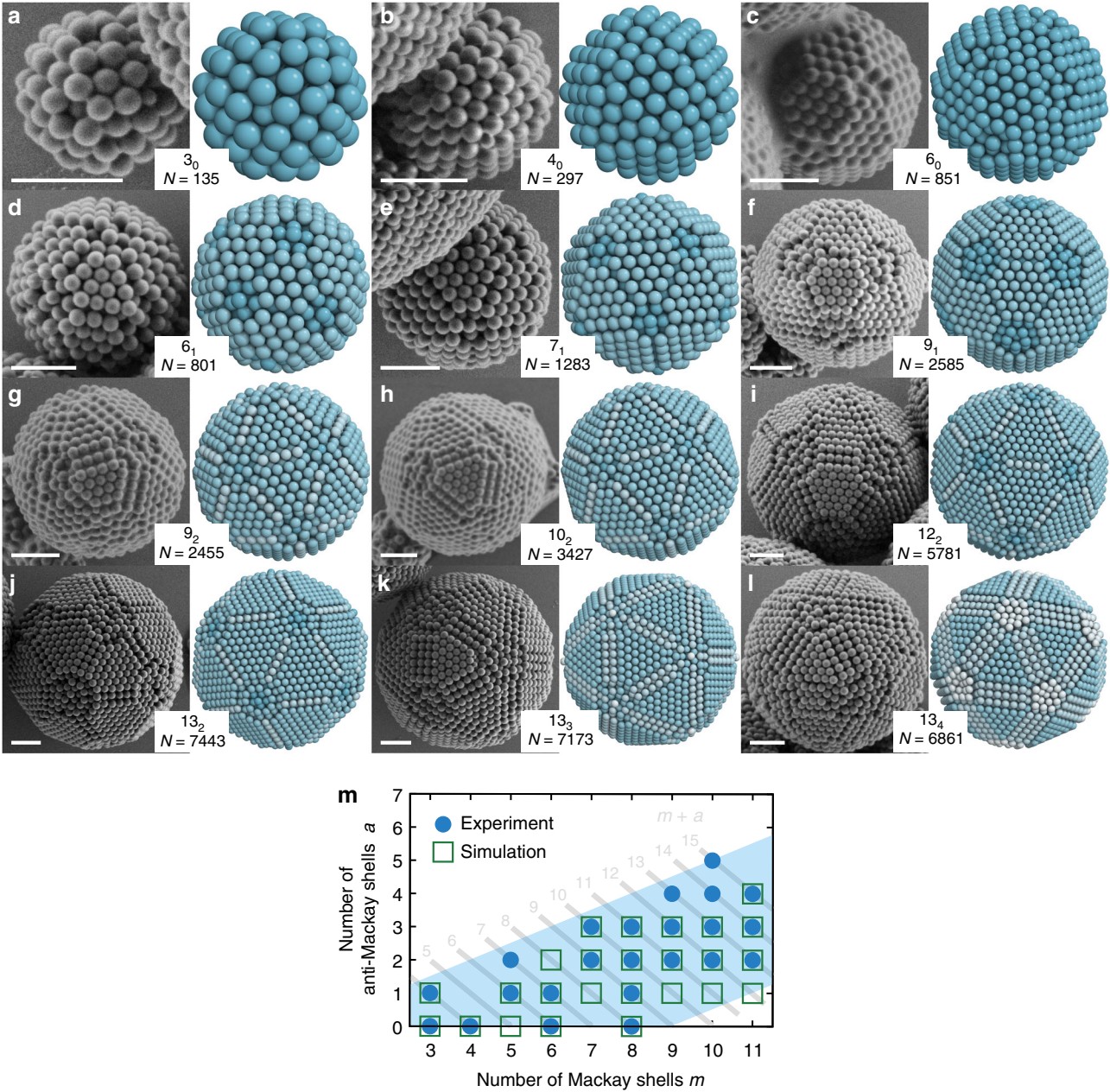

**Fig. 4** Library of magic number colloidal clusters and comparison to model. A rich variety of MCCs are observed with increasing number of particles. **a–c** MCCs without anti-Mackay shells ($m_0$ type) correspond to truncated Mackay icosahedra. **d–f** MCCs with one anti-Mackay shell (($m+1)_1$ type) are characterized by a two particle wide rectangular region and a varying number of Mackay shells. **g–i** Similarly, two anti-Mackay shell clusters (($m+2)_2$ type) feature a width of the rectangular region of three particles. **j–l** MCCs with a fixed total number of 13 shells but a varying number of anti-Mackay shells ($13_a$ type). In each example, SEM images (left) are compared to the corresponding model (right). Scale bars, 1 μm. **m** Summary of all MCCs observed in experiment or simulation organized by type. Clusters with the same total number of shells $m+a$ fall on a diagonal. Such structures are close in the total number of particles. As the cluster size increases, more anti-Mackay shells can typically be accommodated around a larger Mackay core

observations of MCCs in the experiment (Supplementary Figure 7) and simulation (Supplementary Figure 8, see below). The number of anti-Mackay shells increases with cluster size as a result of the improved sphericity but is limited by $a < m/2$.

**Formation and kinetic of MCCs from simulation.** To reveal details of the formation mechanism, we study MCC formation with computer simulations in two steps, as a confocal study to track all particle positions in-situ is difficult due to small particle size and fast diffusion. In the first step ("self-assembly"), colloids in a shrinking droplet during evaporation are modeled as hard

spheres in spherical confinement with decreasing radius using event-driven molecular dynamics (EDMD) simulation (see Methods). This computational approach follows recent work[28] and ignores hydrodynamic interactions, which affect crystallization speed[55,56] and colloidal aggregation far from equilibrium[57,58] but are expected to have a weak influence on the equilibrium cluster structure and near-equilibrium structure formation. We record the equations of state for six system sizes (Fig. 5a, b) and generally observe a sudden pressure drop indicative of a first-order transition from a disordered fluid to an ordered cluster. The second step ("quenching") uses numerical relaxation[59] to mimic the capillary forces that consolidate the

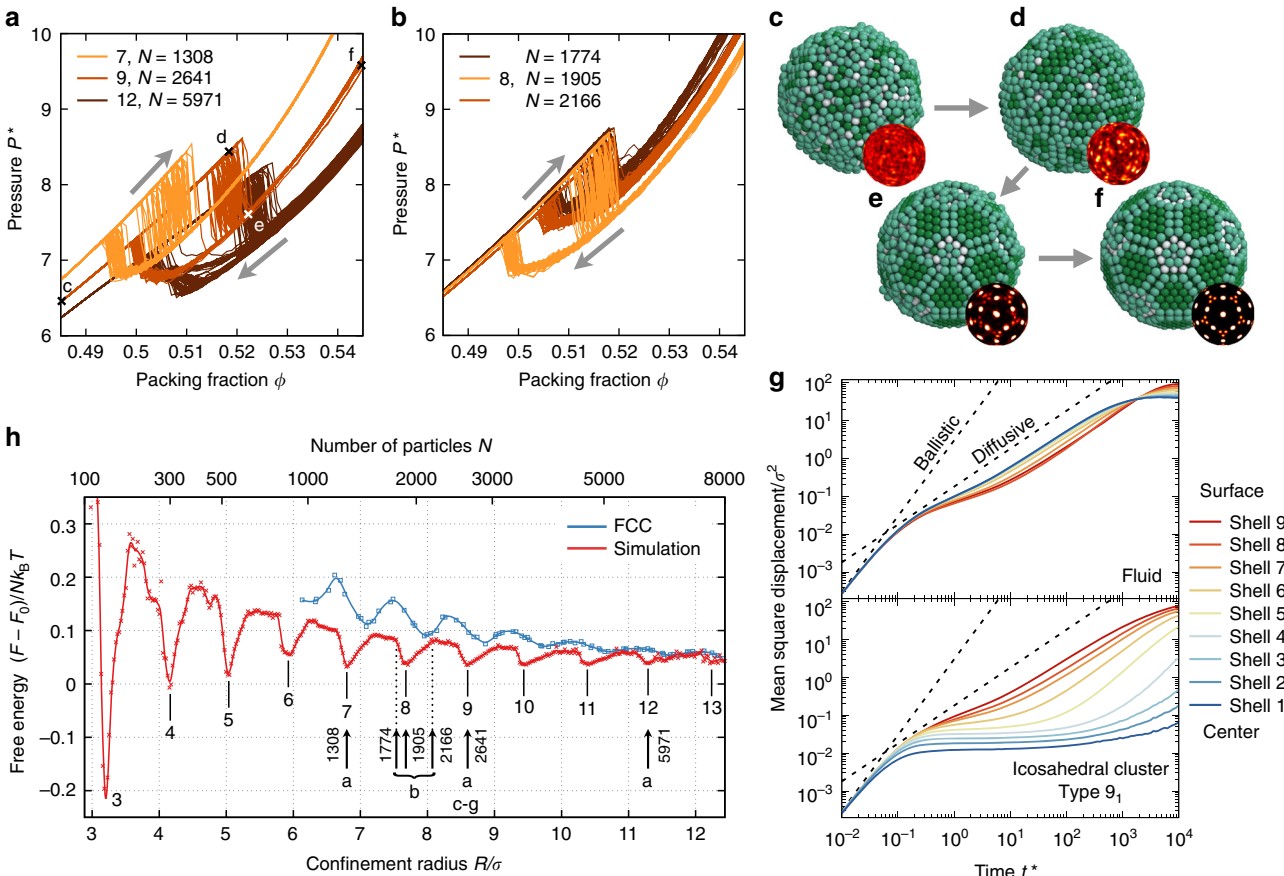

**Fig. 5** Kinetics and stability of magic number colloidal clusters. **a**, **b** Equations of states of hard spheres in spherical confinement. Dimensionless pressure $P^*$ is recorded for sets of 100 simulations. Packing fraction $\phi$ is first increased and then decreased to test for hysteresis. **c**–**f** Snapshots obtained by quenching at the four packing fractions indicated in (**a**). The disordered cluster (**c**) preorders at the surface (**d**) and then rapidly transitions into an icosahedral cluster (**e**). The remaining defects heal given sufficient time (**f**). Particles are colored by the number of neighbors. The bond orientational order diagram (insets) is a global order parameter for the transition from the fluid to the ordered state. **g** Mean square displacement (MSD) of particles for a 9-shell cluster at coexistence packing fraction ($\phi = 0.51$). The dependence of the diffusivity on the shell number reverses at the transition from the fluid (top) to the icosahedral cluster (bottom). The shell number corresponds to the shell the particle was part of at the start of the MSD measurement. **h** Free energy of colloidal clusters computed at $\phi = 0.52$ as a function of the radius of the confining sphere $R$ normalized by subtraction of the free energy of the bulk corrected for surface effects $F_0$ (see Supplementary Information). Free energy minima correspond to clusters with complete shells. Data of clusters with face-centered cubic (FCC) crystal structure is included for comparison for $N > 1000$ where FCC clusters are metastable

colloidal cluster in the final stage of droplet drying where the droplet interface no longer remains spherical due to loss of water volume[60]. In this step, all spheres in the structure obtained from the first step are assigned a Morse pair potential and let to relax into local energy minimum structure. The Morse potential was chosen due to its tunable range and form of interaction. It is important to note that capillary forces and attractive interactions in the quenching step are not essential for the crystallization of hard spheres in confinement. They occur only after the actual self-assembly process and merely push the particles into their final facetted structure but do not change neighbors.

Combining the two steps, the simulations reproduce the transformation from spherical clusters to icosahedral clusters observed in the experiment (Fig. 5c–f). Quenching the ordered clusters of Fig. 5a demonstrates a high propensity for the development of surface features characteristic for MCCs (Supplementary Figures 9-11). However, this ordering process depends sensitively on cluster size. The hysteresis loop becomes wider and pressure in the ordered cluster lower for one cluster size ($N = 1905$ in Fig. 5b) than slightly smaller and larger cluster sizes ($N = 1774$ and $N = 2166$), suggesting size-dependent thermodynamic stability. This will be further elaborated below.

We investigate the kinetics of cluster formation during the assembly process. Particle displacements are recorded over time during EDMD when the packing fraction is increased from 0.48 (fluid phase) to 0.55 (solid phase) and subsequently decreased (Supplementary Figure 12). Due to the presence of hysteresis, the fluid and solid phase occur at the same packing fraction $\phi = 0.51$ during the compression stage and the expansion stage, respectively. In both cases, particles form concentric shells with occasional migration between neighboring shells. To reveal how kinetics varies throughout the cluster, we measure mean square displacement separately for particles starting in different shells. In the fluid phase (Fig. 5g, top), particles near the center diffuse significantly more than particles near the surface, which appear to be hindered in their mobility by the confining wall. In contrast, in the icosahedral cluster (Fig. 5g, bottom), particles in outer shells show an increased mobility while particles in the interior are nearly arrested. This abrupt change in the kinetics is key for understanding MCC formation and highlights the importance of the outermost shells for the ordering process. Crystalline patches develop early in the outer shells and rapidly grow towards the interior[28,41,61]. Importantly, particles near the surface effectively retain mobility through the phase transition. This mobility is

necessary to heal surface defects efficiently and helps forming anti-Mackay surface shells.

**Thermodynamic stability of MCCs.** The predominant formation of closed-shell clusters poses the fundamental question if MCCs are thermodynamically favored. In the absence of particle interactions, we determine free energies of clusters with between 100 and 8000 hard spheres by calculating entropy using the Einstein crystal method. As an important modification of the conventional Einstein crystal method, we include swap moves to sample diffusion efficiently near the ordering transition[62] and subtract the bulk contribution (see Methods and Supplementary Figure 13). The calculated free energy shows a series of distinct minima as the cluster size is varied (Fig. 5h). This demonstrates that icosahedral order is not only favored over FCC for hard spheres in confinement[28] but is realized as thermodynamically stable MCCs with highly defined shell structures (Supplementary Figure 14a). Our analysis reveals that the depth of the free energy wells per particle ranges from 0.6 $k_BT$ for MCCs with 3 shells to 0.018 $k_BT$ for MCCs with 12 shells. The absolute free energy gain for forming a complete shell is constant in the order of 100 $k_BT$ and increases with packing fraction (Supplementary Figure 14b–e). In both experiments and simulations, we observe more anti-Mackay shells as colloidal clusters increases in size (Fig. 4m).

## Discussion

The closed-shell magic number cluster structure requires particles to efficiently utilize space in spherical confinement to maximize entropy, particularly in the vicinity of the curved confinement. Icosahedral symmetry is favored over FCC[28] because the highly facetted closed-shell structure formed by the latter (Wulff shape between octahedron and cube) creates large mismatch and gaps between facets and the spherical confinement. The icosahedral arrangement reduces this mismatch via more but smaller facets, improving local dense packing near the curved interface[63]. Our simulations reveal that there is always a preference for a certain anti-Mackay and Mackay clusters type. The introduction of a few anti-Mackay shells makes the cluster more spherical and compact compared to the truncated Mackay icosahedron (Supplementary Figures 15, 16). This means that the anti-Mackay shells provide means for the cluster to more closely follow the geometry of the confinement and thus use the available space more efficiently. Because the packing in Mackay and anti-Mackay shells differs, both types of shells accommodate different number of particles. Therefore, we speculate that the confined system can adopt different Mackay/anti-Mackay combinations to find the optimal closed-shell magic number arrangements for a given specific number of particles.

The broadened minima in Fig. 5h indicates that magic number clusters can tolerate small deviations in particle numbers. We rationalize this observation from the geometric model. MCCs can adapt different combinations of Mackay and anti-Mackay shells, given a fixed number of total shells, which allows variation in cluster size while maintaining complete shells as detailed above (Fig. 4). Furthermore, we observe a tendency to allocate disorder in the vertex regions (Supplementary Figure 17). Our model predicts that the tetrahedral grains in these regions deviate most strongly from regular FCC packing (Fig. 2d). Therefore, the gain in entropy upon crystallization is lowest in these regions and additional particles can thus be accommodated in the vertex parts with the lowest energy penalties. Additionally, the vertex regions show a corrugated surface termination, which naturally leaves room to accommodate some excess particles. Indeed, magic numbers observed in the simulations are slightly higher than magic numbers acquired from geometric modeling, and the

surplus increases as cluster size increases. When clusters are off-magic number, their free energy is increased. The number of particles forming such clusters is incommensurate with a closed-shell packing with the icosahedral order, producing structural defects in the cluster. These defects lower the entropy of the system by constraining the vibrational degrees of freedom of individual particles[29,63].

The existence of MCCs reported in this work has similarities to other clustering processes, such as micelles[64], fullerene assemblies[65], protein aggregates[66], supramolecules of ionic liquids[67], inorganic clusters[68], and natural framboidal pyrite[69,70]. Just like these natural self-organization processes, MCC formation is efficient in solving the high-dimensional optimization problem of finding a global free energy minimum. Our combined experimental and simulation data suggests that heterogeneous nucleation and subsequent restoration of mobility near the confinement interface, reminiscent of a lubrication layer, is essential for the high structure quality of MCCs. Similar design principles may be applied to hierarchical structures of bimetallic nanoparticles[53], molecular self-assembly[71], and plasmonic nanocluster formation[72]. It remains to be investigated if MCCs can undertake structural transition towards other symmetries as observed in atomic clusters[73], and if magic number effect exists in other finite soft matter systems when the shape[38,74], length scale[75], or rigidity of confinement[76] is varied.

## Methods

**Particle synthesis.** Styrene, acrylic acid, and ammonium peroxodisulfate were purchased from Sigma Aldrich and used as received. PS colloidal particles were synthesized by using acrylic acid as comonomer and ammonium peroxodisulfate as an initiator in surfactant-free emulsion polymerization following literature protocols[77].

**Microfluidics device fabrication.** Microfluidic devices were produced by soft lithography as described in literature[45]. A silicon wafer was spin-coated with negative photoresist SU-8, and patterned by UV light through a photomask. Polydimethylsiloxane (Sylgard 184 PDMS from Dow Corning) was mixed with the curing agent (10:1 ratio) and later poured onto the silicon wafer. Vacuum was applied to degas. The PDMS was cured in the oven at 85 °C overnight. PDMS molds were carefully peeled off and punched with a 0.75 mm diameter biopsy punch to create inlets and outlets. The structured PDMS bonded to a clean glass slide after treatment with oxygen plasma for 18 s at 30 W power. The channels were flooded with Aquapel (PPG Industries) to avoid wetting of water droplets. The device was put in the oven at 85 °C for 1 h for final cleaning.

**Colloidal cluster assembly.** Monodisperse 244 nm PS particles of 1 wt% were suspended in water and loaded into 1 mL syringes (BD disposable syringes). 0.1 wt % perfluoropolypropyleneglycol-block-polyethyleneglycol-block-per-fluor-opolypropyleneglycol surfactant was dissolved in perfluorinated carbon oil (3M Novec Engineering Fluid HFE 7500) PE/2 tubings (0.38 mm/1.09 mm) connect syringes to the microfluidics[45]. Syringe pumps (Cronus) controlled the flow rate of water and the oil phase (50 and 200 μL/h, respectively). Emulsion droplets were collected in 1.5 mL glass vials and sealed with stretched parafilm (Sigma Aldrich) at the opening. Small holes were punched with a 0.4 mm needle (Henke Sass Wolf) into the parafilm to control the speed of evaporation from the vials. Vials were kept in the oven (85 °C), room (25 °C), and fridge (5 °C) for water evaporation.

**Identification of cluster morphologies.** Oil droplets containing colloidal clusters were drop-casted on a silicon wafer for surface pattern examination via scanning electron microscopy (Zeiss Gemini Ultra 50 SEM). As colloidal clusters are randomly deposited on the wafer and are viewed only at the surface from one direction in the SEM, we adopt the following classification scheme. Colloidal clusters without spherical shape were classified as buckled clusters. Clusters with spherical shape and hexagonally close packed surface patterning and with grain boundary scars were classified as spherical clusters. We consider complete local five-fold axes as alternating rings with five complete rectangles and five complete triangles (of possibly varying sizes) at the surface. Those missing any or having under-developed rectangles or triangles are termed incomplete. Only three five-fold axes are visible from one side at most. Clusters with at least one complete local five-fold axis or three incomplete five-fold axes were classified as complete icosahedral clusters. Clusters with one or two local five-fold axes were classified as partially icosahedral clusters. More than 120 colloidal clusters were examined and classified

for each histogram (Fig. 1g, h and Supplementary Figure 1) to achieve statistical significance.

**Electron tomography measurement**. Colloidal clusters dispersed in oil phase were drop-casted directly onto a standard (200 mesh) Lacey carbon copper grid and dried overnight. The grid was mounted in the SEM (FEI Helios NanoLab 660) to identify a suitable structure, orientation, and cluster size for the three-dimensional (3D) analysis. A MCC with a size of 2.8 μm containing about 1200 colloidal particles was selected. The grid with the selected cluster was mounted onto the ultrathin single-tilt tomography holder (Fischione model 2020) and transferred to the transmission electron microscope. STEM tomography was performed using a dual probe- and image-side aberration-corrected FEI Titan[3] Themis 60-300 transmission electron microscope at an acceleration voltage of 300 kV in high-angle annular dark field (HAADF) STEM imaging mode at a camera length of 91 mm. The semi-convergence angle of the STEM probe (microprobe STEM) was reduced to 0.44 mrad to increase the depth of field (DOF) to image all parts of the sample completely in focus throughout the entire tilt series acquisition procedure[78]. The diffraction-limited resolution for the adapted semi-convergence angle was 2.7 nm at a respective DOF of 7.6 μm. The large sample size led to a strong decrease of resolution due to broadening of the STEM probe (multiple elastic scattering)[79], which resulted in an estimated HAADF STEM probe diameter of about 90–120 nm at the bottom surface of the sample in regions with highest projected mass-thickness. The tilt series was acquired using FEI Tomography 4.0 software in a tilt angle range from −76° to 62° with 1° tilt increment, continuous and linear tilting scheme, enabling autofocus and tracking before the acquisition. In order to prevent morphological changes, e.g., shrinkage, of the sample during the electron tomographic tilt series acquisition, a low beam current of 50 pA was applied and the sample was illuminated for 10 min before performing the measurement.

**Electron tomography data analysis**. Tilt series alignment was performed using FEI Inspect 3D software (cross-correlation technique). The tomogram was reconstructed with the simultaneous iterative reconstruction technique (SIRT)[80] over 50 iterations using FEI Inspect 3D software. Reconstructed volumes were visualized with VSG Avizo 8.1 for FEI systems software. A median filter minimized background noise, and a global threshold value was applied to segment the particles from the pore space.

**Model generation**. Clusters were generated by choosing the positions of identical spheres according to geometric construction rules. All models correspond to one or multiple domains with perfect, sheared, or otherwise deformed FCC crystal structure. Deformations were chosen to maximize symmetry and packing fraction. MCCs were constructed in five steps by placing 130 tetrahedral grains, each with $(m+1)(m+2)(m+3)/6$ spheres (Mackay core with $m$ shells). Multiple tetrahedra share spheres at vertices, edges, and faces. The first step follows Mackay[8] by placing 20 tetrahedral grains around a common vertex with the constraint that spheres in each shell touch spheres in the previous shell. In step two, 20 identical tetrahedra were placed over the faces such that the top shells of the Mackay icosahedron acted as mirror planes. In the third step, the gaps were filled by connecting existing vertices. These tetrahedra were slightly more deformed than the previous ones. In the fourth step, vertex regions were filled with pentagonal bipyramids made from five tetrahedral grains such that symmetry was preserved. These tetrahedra had the largest deformation. In the fifth step, all spheres outside of a central sphere with truncation radius $R_{cut}$ were removed. Clusters with FCC structure for the free energy calculations in Fig. 5h were generated by applying only the truncation step to a single FCC grain. The model generation was implemented in a Python code.

**Cluster type classification**. MCC type for clusters assembled in experiment or simulation was identified by evaluating the characteristic rectangular features (width $a + 1$, height $l + 1$) and truncated triangle features (edge lengths $(m − l − a)/2 + 1$ and $l + 1$) with an integer $l$ that describes the amount of vertex truncation. Vertex truncation varies from cluster to cluster (Supplementary Figure 17) broadening the minima in Fig. 5h.

**Self-assembly simulation**. EDMD in NVT mode was implemented in C++ for hard spheres with diameter $\sigma$ and mass $m$ representing colloids. This ansatz builds upon the recent finding[28] that attraction or softness between colloids is not required for icosahedral ordering in emulsion droplets. Collisions of a system consisting of $N$ hard spheres were organized in memory using a tree data structure as priority queue with O(1) complexity[81]. Collisions were handled in a stable fashion[82] such that overlaps that occurred temporarily after compression steps were removed immediately. Spherical confinement was implemented as a hard spherical wall of radius $R$ at packing fraction $\phi = N(\sigma/2R)^3$. Dimensionless units were employed for time $t^\star = t/\tau$ with $\tau = \sigma(m/k_BT)^{1/2}$ and pressure[83]

$$P^* = \frac{P\pi\sigma^3}{6k_BT} = \phi\left(1 + \frac{\sqrt{\pi}}{3}\frac{1}{Nt^\star_{tot}}\left(N_{pc} + \frac{N_{wc}}{\sqrt{2}}\right)\right), \quad (1)$$

where $N_{pc}$ is the total number of particle–particle collisions and $N_{WC}$ the total

number of wall collisions over time $t_{tot}^\star$. Evaporation of the solvent was mimicked by incrementing the packing fraction from $\phi = 0.48$ to $\phi = 0.55$ in steps of 0.001. At each step, a simulation over a duration of $\Delta t^\star = 500$ was performed, which proofed sufficient for robust and reliable icosahedral ordering. Expansion from $\phi = 0.55$ to $\phi = 0.48$ using the same parameters tested for the presence of hysteresis (Fig. 5a, b). Mean square displacement was calculated by temporal averaging after equilibrium had been reached. Particles are assigned to shells according to their starting positions during the measurement and freely mobile throughout the whole cluster.

**Numerical quenching**. The fast inertial relaxation engine (FIRE)[59] implemented in HOOMD-blue[84,85] mimicked capillary forces that consolidate colloidal clusters in the final stage of droplet drying. A force tolerance per particle of $10^{-2}$ and an energy tolerance of $7 \times 10^{-7}$ was used. Hard sphere interaction was replaced with a Morse potential

$$V(r) = D_0\left(e^{-2\alpha(r-r_0)} - 2e^{-\alpha(r-r_0)}\right) \quad (2)$$

with parameters $r_0 = \sigma$, $D_0 = 1$, $a = /\sigma$. Confinement was disabled during quenching.

**Visualization**. Spheres in MCCs generated by the geometric model were colored in shades of blue according to the construction step when the sphere was added. Spheres in colloidal clusters obtained from numerical quenching were colored in shades of green according to the number of nearest neighbors (first peak of the radial distribution function) to highlight surface features.

**Free energy calculation**. Absolute free energy of a colloidal cluster at packing fraction $\phi$ was calculated with the Einstein crystal method[86] applied to hard spheres in hard spherical confinement implemented in a Monte Carlo simulation using harmonic springs, $V(r) = \lambda(r - r_0)^2$. Spring anchor points $\{r_0\}$ were chosen after minimizing the appearance of near-contacts between spheres via rapid compression of the colloidal cluster to high density and re-expansion to $\phi$. The spring constant $\lambda$ was increased logarithmically in discrete steps over the range $10^{-5}$ to $10^5$. Particle swap moves[62] sped up diffusion processes. At each step for $\lambda$ the averaging was performed long enough for relaxation. The calculation for Fig. 5h was repeated five times and averaged to decrease numerical error. Spheres were assigned to a shell $s$ in a cluster of $S$ shells by distance to the center via $s = [(R − r)(S − 0.5)/R]$. The average shell thickness in the packing fractions simulated with EDMD was $0.95\sigma$. Absolute free energy values were reported after subtracting the bulk contribution $F_0(N,\phi)$. Starting points for the free energy calculation are equilibrated simulation snapshots and constructed ideal FCC, Mackay, and anti-Mackay cluster. We are aware that the constructed clusters are not equilibrated. For this reason, the free energies are only upper limits in the case of the constructed clusters.

**Free energy bulk contribution**. The entropy of a finite system of $N$ hard spheres in the FCC crystal structure was expressed using free volume theory[87],

$$\frac{S_{FCC}(N,\phi)}{Nk_B} = s_0 + \log(N) + f\log\left(\frac{\phi_{max}}{\phi} - 1\right) \quad (3)$$

with the FCC jamming limit (maximal packing fraction) $\phi_{max} = \pi/\sqrt{18}$ and the effective number of degrees of freedom per particle $f = 3$. The constant $s_0$ was obtained from a fit to EDMD simulations of hard spheres with periodic boundaries as $s_0 = 12.57$ (Supplementary Figure 13). The bulk contribution to the free energy of colloidal clusters was $F_0(N,\phi) = −TS_{FCC}(N,\phi_c)$, where the packing fraction $\phi_c(\sigma_c) = N(\sigma/D_c)^3$ was calculated with a confinement sphere diameter

$$D_c = 2R_c = \sigma\left(\frac{N}{\phi}\right)^{\frac{1}{3}} - D_0 \quad (4)$$

that was corrected empirically by subtracting a constant $D_0$ in order to take into account the effect of confinement. In Fig. 5h and Supplementary Figure 14a, we chose $D_0/\sigma = 0.4$. In Supplementary Figure 14b–d, we chose $D_0/\sigma = 0.436, 0.486, 0.520$ for $\phi = 0.52, 0.55, 0.57$, respectively. The effect of confinement increased with packing fraction.

## Data availability

Data generated during the current study are available from the corresponding authors on reasonable request. A Python 3 code "generateMCC.py" that generates MCC models following the procedure in Fig. 2b using the XYZ file format is included as Supplementary Information. Custom EDMD simulation code used in the current study is available from the corresponding authors on reasonable request.

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

## Acknowledgements

This work was supported by Deutsche Forschungsgemeinschaft (DFG) through the projects EN 905/2-1 and VO 1824/7-1 to M.E. and N.V., respectively. T.P. acknowledges support by DFG within the framework of the research training group GRK 1896. All authors acknowledge the Cluster of Excellence Engineering of Advanced Materials Grant EXC 315/2, the Central Institute for Scientific Computing (ZISC), the Interdisciplinary Center for Functional Particle Systems (FPS), and the Center for Nanoanalysis and Electron Microscopy (CENEM) at Friedrich-Alexander University Erlangen-Nürnberg. Computational resources and support provided by the Erlangen Regional Computing Center (RRZE) are gratefully acknowledged.

## Author contributions

J.W. performed microfluidics experiments, assembled and characterized the colloidal clusters, and proposed the model; C.F.M. performed the event-driven simulations; M.E. wrote the event-driven simulation code and calculated free energies; T.P. and B.A.Z. performed electron microscopy and tomographic reconstruction; J.W., C.F.M., M.E., and N.V. analyzed the data from experiment and simulation; J.W., C.F.M., M.E., and N.V. wrote the manuscript; all authors discussed the results and contributed to the final version of the manuscript; E.S., M.E., and N.V. supervised the research; M.E. and N.V. conceived and directed the study.

## Additional information

**Competing interests:** The authors declare no competing interests.

