## [Peer Review File · Nature Communications]

Reviewers' comments:

Reviewer #1 (Remarks to the Author):

This paper contains an impressive demonstration of the possibility of producing large high-symmetry colloidal clusters of the icosahedral family in confined systems. Calculations support the conclusion that these structures are indeed equilibrium ones, being favoured by a larger entropy than other structures. Complex surface arrangements including anti-Mackay shells are found.

I think that the results of this paper are very important, so that I recommend publication in Nature Communications. However, some revision is necessary, as explained in the points below.

Main point. The authors should better explain the origin of the entropy differences for the following cases:

- a) Magic numbers vs non-magic numbers. The latter present defects. In atomic clusters, energy is the driving force for magic structures, while non-magic structures may prevail because of their higher entropy (both configurational and vibrational). What is the driving force here? Why defects are not beneficial in terms of entropy?
- b) Icosahedral vs fcc packing. The origin of entropy difference in this case may be already discussed in the literature. Anyway, key points should be recalled.
- c) Mackay vs anti-Mackay. The authors state that a few anti-Mackay shells can indeed produce a denser packing because they make the clusters more spherical. I understand that sphericity is beneficial when discussing energetic stability (mostly because of the maximization of nearest-neighbour bonds) but why should it be beneficial in terms of entropy?

Minor points:

- The sentence "In contrast to atomic clusters that can minimize free energy by addition or removal of atoms to reach magic numbers" is misleading since in several concerning atomic clusters in which such processes are unlikely, and cluster structures are dominated by kinetic effects, especially when sizes exceeds a few ten atoms (see for example D. M Wells et al, *Nanoscale* 7, 6498-6503 (2015)).
- The structure in Fig. 2 resulting from the addition of 60 tetrahedra is a pentakis dodecahedron. Pentakis-dodecahedral shapes have been proposed for binary metallic clusters by DFT calculations (see K. Laasonen et al., *J. Phys. Chem C* 117, 26405-26413 (2013)).

Reviewer #2 (Remarks to the Author):

This manuscript describes the observation of clusters with a magic number of colloidal spheres self-assembled in the spherical confinement of an emulsion droplet as studied in experiments as well as in simulations. The authors show using free-energy calculations that these magic number clusters with Mackay and anti-Mackay shells correspond to minimum free-energy clusters compared to clusters with a slightly larger or smaller number of spheres. The Mackay and anti-Mackay clusters were already found by de Nijs et al. *Nature Materials* 2015, and as such this work can be seen as incremental to that work. However, the authors performed a very thorough analysis of cluster sizes in the range of 100-7000 spheres and identified very carefully the number of Mackay and anti-Mackay shells for each cluster by analysing the surface pattern of the clusters. Unfortunately, the mechanism of the formation of Mackay versus anti-Mackay shells is still not clear to me. What determines if a cluster favors the formation of a Mackay shell or an anti-Mackay shell? Is this solely determined by the number of particles? This requires a better explanation or investigation? There are a couple of other places in the manuscript that needs further clarification.

In the abstract: the sentence "we show that magic number clusters exist as local free energy minima

in discrete systems without attractive interactions, where entropy maximization governs structure formation." I find this sentence misleading as spherical confinement is also required to obtain these clusters. The same hold for the sentence "Our findings demonstrate that the concept of magic numbers is not limited by the types of interactions...." This is only true when the cluster is hold together by confinement. Also the concluding sentence: "Magic states may thus compete with bulk symmetries and direct the structure in self-organizing systems across the scales". This holds only for finite clusters. For cluster sizes larger than about 100.000 bulk-like behaviour is more favourable as shown in de Nijs, Nature Materials (2015).

On page 2: Small number of colloids pack into defined clusters.... Perhaps the authors can also mention the work of I. Schwarz, J. Chem. Phys. 135, 244501 (2011) and B. Peng, Angewandte Chemie International Edition 52, 6709-6712 (2013). And again this is only in confinement.

On page 2: I am missing details on the experimental system. For instance, what kind of colloids are used, are they charged or sterically stabilized? what is the size of the particles, the solvents, interactions of the particles, etc.? These details might be boring but important for the analysis that was done. For instance, do you need electron microscopy or can you use confocal? Can you track the particle positions, etc.?

On page 3: I don't understand the sentence " Finally, we apply spherical truncation to mimic the confinement of the emulsion droplet". What is actually meant here? Please, rephrase.

On page 3: By comparison with the model, we can now identify the MCC in Fig. 2a as of 12_2 type, ... Do the authors mean that there is a unique correspondence between the surface pattern and the number of Mackay and anti-Mackay shells? If this is true than perhaps it is worth stressing it more clearly.

On page 4: the icosahedral cluster shows two-, three-, and five-fold symmetry. Does this mean that the cluster can also be called quasicrystalline?

On page 4: ..., two triangles of 21 hexagonally close-packed particles.... Fig. 2h,I indicates a triangle and a pentagon and not two triangles, which is misleading. ...Four particles at the surface.... (indicated by a blue line)... it is not clear where this blue line is, perhaps the rectangle in Fig. 2f,g??

On page 4: The simulation protocol for the second step that includes capillary forces is not very clear in the main manuscript. Perhaps, the authors can explain what they did and what they try to mimic. It would help the reader if some of the details is moved from the supplementary info to the main manuscript. What kind of simulations are performed? Monte Carlo or Molecular dynamics? What are the interactions between the particles, etc.? Reading the supplementary info on the second step: It seems that the capillary forces are mimicked by the Morse potential. This needs further explanation. Why the Morse potential and the mentioned parameters for the Morse potential? Capillary forces are extremely long-ranged and have a logarithmic form. Do all the particles interact with a Morse potential or only particles attached at the retracting interface for which we expect that capillary forces will play a role? Do the number of particles always correspond to only the magic numbers? If not, what is done with the excess number of particles? Do they also interact with the Morse potential? In the experiments the loose particles may fall off

On page 5: the kinetics is investigated by calculating the mean square displacement of the particles. Is this the mean square displacement within a shell? What happens with particles that move out of a shell? Or how is the mean square displacement calculated? Please, give more details.

On page 5: free-energy calculations were performed for 100-8000 spheres in a spherical confinement. What is done with the empty spaces in the spherical confinement in the case of icosahedral clusters? If the harmonic springs become very weak the outer shells can melt or does this not occur? In principle, extra spheres can be added to fill these voids. Or is the cluster still hold together by the Morse potential?

In the supplementary info, the authors classified clusters with at least one complete local five-fold patterns or three incomplete five-fold patterns as complete icosahedral clusters. What do the authors mean? A complete icosahedral cluster should have 12 five-fold patterns. Only one of the 12 is a very small number. How do the results change if a complete icosahedral cluster should have 12 five-fold

patterns? Or is this extremely rare? What do the authors mean by complete? The surface pattern should have five-fold symmetry or the 5 truncated triangular and 5 rectangular surface patterns should be complete or the 5 3-dimensional tetrahedral grains should be complete? Please, specify. Supplementary info page 4: The pressure is defined in terms of the particle-particle collisions and particle-wall collisions. As the system is inhomogeneous the pressure should be replaced by a pressure tensor that depends on position. Did the authors calculate the pressure tensor as a function of the position? How homogeneous and isotropic is the pressure in the cluster?

Reviewer #3 (Remarks to the Author):

Report for Magic Number Colloidal Clusters as Minimum Free Energy Structures by Wang et al.

The authors use microfluidic techniques to bind nanoparticles into emulsion droplets, which are subsequently evaporated to yield clusters of nanoparticles. Using computer simulation, they find that such particles with excluded volume interactions exhibit “magic numbers” in that certain cluster geometries are favoured from the point of view of the free energy. The work is potentially interesting, though there is some considerable way to go before publication can be considered.

It appears to me upon reading the manuscript that there are really two sides to this story. On the one hand are experiments. But apart from demonstrating the existence of Mackay and anti-Mackay icosahedra, I see little that is really new. Mainly this is because ref 17 already demonstrates the experimental technique. While Ref 17 reports icosahedral clusters, and seems to me truly groundbreaking (enough to get a mention on the “Big Bang theory” TV show), it does not discuss magic numbers. So the main development of this paper is magic number clusters.

But those are only shown with simulation. Again, ref 17 used simulation to determine the minimum free energy clusters.

In short, the authors have used established methods to infer that Mackay and anti-Mackay icosahedra are magic number clusters. Given the extensive literature on atomic clusters (e.g. Baletto and Ferando Rev. Mod. Phys. 77 371 (2005)), which I think the authors should cite, this isn't really a big surprise. While indeed the authors have a confined system, rather than attractive atoms, the similarity in structure between the (admittedly smaller) colloidal clusters made in the pioneering work on such confined systems of Manoharan (ref 14) and atomic clusters already hints strongly that magic numbers should also be seen for nanoparticle/colloidal clusters. Indeed, such a result seems unavoidable from the famous WCA paper (J. Chem. Phys. 54, 5237 (1971)).

The authors use an excessive amount of hyperbole in their manuscript. One example (there are more!) is that the statement “Our findings demonstrate that the concept of magic numbers is not limited by the types of interactions and much more ubiquitous than previously believed” badly needs to be toned down. One more interaction potential (hard sphere) on top of the very many existing, for example in the Cambridge Cluster database, can hardly be grounds for a “much” higher level of ubiquity. Magic numbers in the sense of shells which the authors seem to use aren't exactly ubiquitous anyway, clusters of carbon being an obvious example!

The authors could help their case, in terms of the generality of their work by citing C60 clusters, which also exhibit the kind of magic numbers they find with the nanoparticles - see, again, Baleto and Ferando *Rev. Mod. Phys.* 77 371 (2005).

In summary, the authors need to tone down their manuscript and be very much more upfront about what they have actually achieved. That they have used simulation to predict magic number clusters in colloidal/nanoparticle systems is interesting, and that they have observed such clusters experimentally is noteworthy. They do not seem to show any statistics in their experiments: I realise that is hard with this technique, but something along the lines of the breakthrough work of Meng et al. (*Science* 327 560 (2010)) would help their case considerably.

Reviewer 1

This paper contains an impressive demonstration of the possibility of producing large high-symmetry colloidal clusters of the icosahedral family in confined systems. Calculations support the conclusion that these structures are indeed equilibrium ones, being favoured by a larger entropy than other structures. Complex surface arrangements including anti-Mackay shells are found.

I think that the results of this paper are very important, so that I recommend publication in Nature Communications. However, some revision is necessary, as explained in the points below.

We thank the reviewer for the positive recognition of our results. We revised the manuscript as described below.

Main point. The authors should better explain the origin of the entropy differences for the following cases:
a) Magic numbers vs non-magic numbers. The latter present defects. In atomic clusters, energy is the driving force for magic structures, while non-magic structures may prevail because of their higher entropy (both configurational and vibrational). What is the driving force here? Why defects are not beneficial in terms of entropy?

Entropy is the only driving force in our hard particle system. The reviewer is correct that defects are associated with a slight increase in configurational entropy. After all there are seemingly endless ways to create and distribute defects compared to the unique magic number clusters. However, defects also decrease packing efficiency of the colloids in the cluster and thus decrease vibrational entropy. In well-formed magic clusters, vibrational effects are the dominant factor above the critical packing fraction where the ordered cluster is thermodynamically stable. The situation is comparable to the observation of perfect FCC crystals in the bulk. FCC defects (vacancies, interstitials, dislocations, stacking faults etc.) are associated with entropy penalties. They have a low probability of remaining in the crystal long after nucleation. Magic clusters correspond to deep wells in the free energy (negative entropy) landscape, which are difficult to disturb by defects. In contrast, non-magic clusters correspond to shallow minima that are often accompanied by defects. After all, the situation is not so different from atomic clusters where energy is the driving force.

Our significantly revised introduction and discussion now contrast the situation in atomic clusters and colloidal clusters more clearly. We added in the discussion section:

“When clusters are off-magic number, their free energy is increased. The number of particles forming such clusters is incommensurate with a closed-shell packing in an icosahedral crystal, producing defects in the crystal arrangement. These defects lower the entropy of the system by constraining the vibrational degrees of freedom of the individual particles (12, 54).”

b) Icosahedral vs fcc packing. The origin of entropy difference in this case may be already discussed in the literature. Anyway, key points should be recalled.

c) Mackay vs anti-Mackay. The authors state that a few anti-Mackay shells can indeed produce a denser packing because they make the clusters more spherical. I understand that sphericity is beneficial when

discussing energetic stability (mostly because of the maximization of nearest-neighbour bonds) but why should it be beneficial in terms of entropy?

In response to these points, we included the following paragraph at the beginning of the new discussion section:

“The closed-shell magic cluster structure requires particles to efficiently utilize space in spherical confinement to maximize entropy, particularly in the vicinity of the curved confinement. Icosahedral symmetry is favored over FCC (22) because the highly faceted closed-shell structure formed by the latter (Wulff shape between octahedron and cube) creates large mismatch and gaps between facets and the spherical confinement. Icosahedral arrangement reduces this mismatch via more but smaller facets, creating better local dense packing near the curved interface (54). Our simulations reveal that although anti-Mackay and Mackay clusters can easily convert into another if the packing fraction is near the critical packing fraction, there is always a preference for a certain cluster type. The introduction of a few anti-Mackay shells makes the cluster more spherical and compact compared to the truncated Mackay icosahedron (Supplementary Fig. S15, S16). This means that the anti-Mackay shells provide means for the cluster to more closely follow the geometry of the confinement and thus use the available space more efficiently. Because the packing in Mackay and anti-Mackay shells differs, both types of shells accommodate different number of particles. Therefore, we speculate that the confined system can adopt different Mackay/anti-Mackay combinations to find the optimal closed-shell magic number arrangements for a given specific numbers of particles.”

Minor points:

- The sentence "In contrast to atomic clusters that can minimize free energy by addition or removal of atoms to reach magic numbers" is misleading since in several concerning atomic clusters in which such processes are unlikely, and cluster structures are dominated by kinetic effects, especially when sizes exceeds a few ten atoms (see for example D. M Wells et al, Nanoscale 7, 6498-6503 (2015)).

We thank the reviewer for the reference, which we have cited in the main text now (Ref. 64). We modified the sentence as follows at the end of the introduction:

“While the dominant occurrence of atomic clusters with magic numbers in mass spectra (3, 4) suggests self-adjustment in their sizes during formation by addition or removal of atoms to reach magic numbers under appropriate circumstances, the number of particles in a colloidal cluster remains fixed in confinement, which allows experiments to explore clusters of arbitrary sizes.”

and included the following text at the end of the conclusion:

“It remains to be investigated if MCCs can undertake structural transition towards other symmetries as observed in atomic clusters (64), and if magic number effect exists in other finite soft matter systems when the shape (33, 65), length scale (66), or rigidity of confinement (67) is varied.”

- The structure in Fig. 2 resulting from the addition of 60 tetrahedra is a pentakis dodecahedron. Pentakis-dodecahedral shapes have been proposed for binary metallic clusters by DFT calculations (see K. Laasonen et al., J. Phys. Chem C 117, 26405-26413 (2013)).

We thank the reviewer for identifying the polyhedron and the reference, which has been added to Ref. 46. We mentioned the Pentakis dodecahedron in the caption of Figure 2 and amended the text as follows:

“Our model extends the well-established Mackay icosahedron (47) to a new sphere packing model into a Pentakis dodecahedron, which may help the understanding of binary clusters where slightly larger component resides in more deformed regions (46, 48, 49).”

Reviewer 2

This manuscript describes the observation of clusters with a magic number of colloidal spheres self-assembled in the spherical confinement of an emulsion droplet as studied in experiments as well as in simulations. The authors show using free-energy calculations that these magic number clusters with Mackay and anti-Mackay shells correspond to minimum free-energy clusters compared to clusters with a slightly larger or smaller number of spheres. The Mackay and anti-Mackay clusters were already found by de Nijs et al. Nature Materials 2015, and as such this work can be seen as incremental to that work. However, the authors performed a very thorough analysis of cluster sizes in the range of 100-7000 spheres and identified very carefully the number of Mackay and anti-Mackay shells for each cluster by analysing the surface pattern of the clusters.

We thank the reviewer for acknowledging our efforts of identifying and analyzing the precise structure of magic number colloidal clusters.

Unfortunately, the mechanism of the formation of Mackay versus anti-Mackay shells is still not clear to me. What determines if a cluster favors the formation of a Mackay shell or an anti-Mackay shell? Is this solely determined by the number of particles? This requires a better explanation or investigation?

The preference of anti-Mackay over Mackay cluster is indeed solely determined by the number of particles. Mackay and anti-Mackay clusters slightly differ in their number of particles (Fig. 4). We observed a general trend that more anti-Mackay shells are favored as cluster size increases within a well associated with a magic number in Fig. 5h due to their higher compactness and the related more efficient local packing near the spherical interface. We extended the manuscript as follows at the beginning of the discussion:

“Our simulations reveal that although anti-Mackay and Mackay clusters can easily convert into another if the packing fraction is near the critical packing fraction, there is always a preference for a certain cluster type. The introduction of a few anti-Mackay shells makes the cluster more spherical and compact compared to the truncated Mackay icosahedron (Supplementary Fig. S15,S16). This means that the anti-Mackay shells provide means for the cluster to more closely follow the geometry of the confinement and thus use the available space more efficiently. Because the packing in Mackay and anti-Mackay shells differs, both types of shells accommodate different number of particles. Therefore, we speculate that the confined system can adopt different Mackay/anti-Mackay combinations to find the optimal closed-shell magic number arrangements for a given specific numbers of particles.”

There are a couple of other places in the manuscript that needs further clarification. In the abstract: the sentence “we show that magic number clusters exist as local free energy minima in discrete systems without attractive interactions, where entropy maximization governs structure formation.” I find this sentence misleading as spherical confinement is also required to obtain these clusters. The same hold for the sentence “Our findings demonstrate that the concept of magic numbers is not limited by the types of interactions....” This is only true when the cluster is hold together by confinement.

Also the concluding sentence: “Magic states may thus compete with bulk symmetries and direct the structure in self-organizing systems across the scales”. This holds only for finite clusters. For cluster sizes larger than about 100.000 bulk-like behaviour is more favourable as shown in de Nijs, Nature Materials (2015).

We thank the reviewer for these comments have revised the abstract to:

“To this point, magic number clusters have been exclusively found with attractive interactions as present between atoms. Here we show that magic number clusters exist in a confined soft matter system. [...] Targeting similar magic number states is a strategy towards unique configurations in finite self-organizing systems across the scales.”

On page 2: Small number of colloids pack into defined clusters.... Perhaps the authors can also mention the work of I. Schwarz, J. Chem. Phys. 135, 244501 (2011) and B. Peng, Angewandte Chemie International Edition 52, 6709-6712 (2013). And again this is only in confinement.

We thank the reviewer for providing these references and added them as Ref. 31 and Ref. 32.

On page 2: I am missing details on the experimental system. For instance, what kind of colloids are used, are they charged or sterically stabilized? what is the size of the particles, the solvents, interactions of the particles, etc.? These details might be boring but important for the analysis that was done. For instance, do you need electron microscopy or can you use confocal? Can you track the particle positions, etc.?

We added more experimental details to the main text:

“We study the system size-dependent colloidal cluster formation in monodispersed droplets of an aqueous dispersion of polystyrene (PS) colloidal particles with different concentration in a continuous oil phase produced by microfluidics (Supplementary Fig. S1). The PS colloidal particles are 244 nm in diameter and stabilized by carboxylate surface groups introduced as comonomers in the synthesis by surfactant-free emulsion polymerization. [...] To reveal details of the formation mechanism, we study MCC formation with computer simulations in two steps, as a confocal study to track all particle positions in-situ is difficult due to small particle size and fast diffusion.”

Additionally, the entire experimental section is now included in the main text body to facilitate reproduction and follow-up experiments by interested researchers.

On page 3: I don't understand the sentence ” Finally, we apply spherical truncation to mimic the confinement of the emulsion droplet”. What is actually meant here? Please, rephrase.

We constructed a geometric model for the arrangement of the particles in the cluster. But as the model has sharp vertices and is not compatible with the spherical droplet interface, a spherical truncation to remove spheres within the vertices is necessary to accurately reproduce the observed cluster structure. We added a more detailed explanation on page 4:

“Finally, we apply spherical truncation to remove spheres in the model whose distances to the center is larger than the truncation radius. This mimics the effect of confinement of the confined colloids, which are forced into a spherical shape by the geometry of the emulsion droplet.”

On page 3: By comparison with the model, we can now identify the MCC in Fig. 2a as of 12_2 type, ... Do the authors mean that there is a unique correspondence between the surface pattern and the number of Mackay and anti-Mackay shells? If this is true than perhaps it is worth stressing it more clearly.

Yes, there is a unique correspondence. It is possible to identify the interior shell structure from the surface pattern. We extended the following paragraph:

“The surface is tiled with rectangles and truncated triangles, which alternate around five-fold axes. Together, these structural elements form a closed shell spanning the cluster’s surface (Fig. 2a). [...] The distinct surface patterns of the MCCs are directly correlated with their crystal structure and can be used to deduce the number of Mackay and anti-Mackay shells. From the model, we identify that the rectangular surface features are the characteristic structural element associated with the anti-Mackay shells (Fig. 2b). They result from truncation of the 20 twinned tetrahedra added to the facets of the icosahedral structure and the additional 90 tetrahedra filling the open voids of the model structure. In the absence of an anti-Mackay shell, a single line of colloids marks the facet of the icosahedron. A single anti-Mackay shell produces a rectangle with a width of two particles, two anti-Mackay shells a width of three particles (as in the example structure) and so on.”

On page 4: the icosahedral cluster shows two-, three-, and five-fold symmetry. Does this mean that the cluster can also be called quasicrystalline?

The icosahedral symmetry originates from 20 twinned tetrahedra with a slightly deformed fcc lattice. Therefore, it is not related to quasicrystallinity. We clarified in the main text body:

“Note that the icosahedral order of the model and MCCs originates from multi-twinned tetrahedral grains with a deformed face cubic center crystal structure, which bears no direct connection to quasicrystals.”

On page 4: ..., two triangles of 21 hexagonally close-packed particles.... Fig. 2h,I indicates a triangle and a pentagon and not two triangles, which is misleading. ...Four particles at the surface.... (indicated by a blue line)... it is not clear where this blue line is, perhaps the rectangle in Fig. 2f,g??

We thank the reviewer for pointing out the missing triangle. We revised the figure. The mentioned figure is now Fig. 3. The blue line was replaced with a green line for better visibility.

On page 4: The simulation protocol for the second step that includes capillary forces is not very clear in the main manuscript. Perhaps, the authors can explain what they did and what they try to mimic. It would help the reader if some of the details is moved from the supplementary info to the main manuscript. What kind of simulations are performed? Monte Carlo or Molecular dynamics? What are the interactions between the particles, etc.?

To clarify our procedure, we moved some details of the simulation procedure from supplementary information to the main text. We also added all experimental details from the supporting information to the methods section in the main text body.

“The second step (“quenching”) uses numerical relaxation (50) to mimic the capillary forces that consolidate the colloidal cluster in the final stage of droplet drying where the droplet interface no longer remains spherical due to loss of water volume (51). In this step, all spheres in the structure obtained from the first step are assigned a Morse pair potential and let to relax into local energy minimum structure. The Morse potential was chosen due to its tunable range and form of

interaction. It is important to note that capillary forces and attractive interactions in the quenching step are not essential for the crystallization of hard spheres in confinement. They occur only after the actual self-assembly process and merely push the particles into their final faceted structure but do not change neighbors.”

Reading the supplementary info on the second step: It seems that the capillary forces are mimicked by the Morse potential. This needs further explanation. Why the Morse potential and the mentioned parameters for the Morse potential?

Capillary forces are extremely long-ranged and have a logarithmic form. Do all the particles interact with a Morse potential or only particles attached at the retracting interface for which we expect that capillary forces will play a role?

The choice of the Morse potential is not critical. The Morse potential has the advantage over the Lennard-Jones potential of allowing to vary the well width. During the second step particles only relax locally. All particles feel attractive interactions with their neighbors. Confinement was disabled during quenching. Our motivation of its use is to “make the clusters look prettier in pictures”. Besides Fig. 5c-f, only the Supplementary Fig. S8-S11 make use of it. No scientific conclusions of the manuscript rely on the use of the Morse potential.

Do the number of particles always correspond to only the magic numbers? If not, what is done with the excess number of particles? Do they also interact with the Morse potential? In the experiments the loose particles may fall off.

We quench all final configurations of EDMD simulations. The number of particles can be arbitrary (we vary the numbers systematically) including magic numbers and off-magic numbers. All particles interact with the Morse potential. In experiments, the PS particles (244 nm) are held together by van der Waals forces after drying. While we cannot exclude that particles fall off the surface, we have not seen any experimental evidence for such behavior.

On page 5: the kinetics is investigated by calculating the mean square displacement of the particles. Is this the mean square displacement within a shell? What happens with particles that move out of a shell? Or how is the mean square displacement calculated? Please, give more details.

We updated the manuscript text:

“We investigate the kinetics of cluster formation during the assembly process. Particle displacements are recorded over time during EDMD simulation when the packing fraction is increased from 0.48 (fluid phase) to 0.55 (solid phase) and subsequently decreased (Supplementary Fig. S12). Due to the presence of hysteresis, the fluid and solid phase occur at the same packing fraction $\phi = 0.51$ during the compression stage and the expansion stage, respectively. In both cases, particles form concentric shells with occasional migration between neighboring shells. To reveal how kinetics varies throughout the cluster, we measure mean square displacement separately for particles starting in different shells.”

and added the following text to Methods:

“Mean square displacement was calculated by temporal averaging after equilibrium had been reached. Particles are assigned to shells according to their starting positions during the measurement and freely mobile throughout the whole cluster.”

On page 5: free-energy calculations were performed for 100-8000 spheres in a spherical confinement. What is done with the empty spaces in the spherical confinement in the case of icosahedral clusters? If the harmonic springs become very weak the outer shells can melt or does this not occur? In principle, extra spheres can be added to fill these voids. Or is the cluster still hold together by the Morse potential?

The clusters adopt truncated icosahedral (Mackay clusters) or involve anti-Mackay shells (anti-Mackay clusters) to efficiently pack and fill in the empty space in the spherical confinement, which is purely stabilized by entropy. No Morse potential is used or necessary in for the structure formation (see above). Indeed, the outer shell does melt and rearrange, which we discuss in the context of Figure 5. This two-step crystallization mechanism is especially important for the formation of the highly regular anti-Mackay shell. No extra spheres are added during the crystallization. The system is closed and maintains an identical number of particles.

Performing free energy calculations in the presence of rearrangements in the outer shells is a central challenge of the numerical work for which we had to rely particle swaps (Schilling et al., Ref. 53), see the Methods for details.

In the supplementary info, the authors classified clusters with at least one complete local five-fold patterns or three incomplete five-fold patterns as complete icosahedral clusters. What do the authors mean? A complete icosahedral cluster should have 12 five-fold patterns. Only one of the 12 is a very small number. How do the results change if a complete icosahedral cluster should have 12 five-fold patterns? Or is this extremely rare? What do the authors mean by complete? The surface pattern should have five-fold symmetry or the 5 truncated triangular and 5 rectangular surface patterns should be complete or the 5 3-dimensional tetrahedral grains should be complete? Please, specify.

We chose this classification because of experimental limitations in the statistical evaluation. We use an SEM to characterize the particles. Statistics on tomography are not possible because of the time consuming procedure. The colloidal clusters are randomly deposited on the substrate, which can only be observed in one perspective in SEM. At maximum only three five-fold axes are visible. A few exemplary tilting tests demonstrated that the other facets are generally well-formed if sufficiently many five-fold axes are present on one side. Complete icosahedral clusters are very common in our experiments when the right conditions are met and the drying process is slowed down sufficiently. We added more details to the methods:

“As colloidal clusters are randomly deposited on the wafer and are viewed only at the surface from one direction in the SEM, we adopt the following classification scheme. Colloidal clusters without spherical shape were classified as buckled clusters. Clusters with spherical shape and hexagonally close packed surface patterning and with grain boundary scars were classified as spherical clusters. We consider complete local five-fold axes as alternating rings with five complete rectangles and five complete triangles (of possibly varying sizes) at the surface. Those missing any or having under-developed rectangles or triangles are termed incomplete. Only three five-fold axes are visible from one side at most. Clusters with at least one complete local five-fold axis or three incomplete five-fold axes were classified as complete icosahedral clusters. Clusters with one or two local five-fold axes were classified as partially icosahedral clusters.”

Supplementary info page 4: The pressure is defined in terms of the particle-particle collisions and particle-wall collisions. As the system is inhomogeneous the pressure should be replaced by a pressure tensor that depends on position. Did the authors calculate the pressure tensor as a function of the position? How homogeneous and isotropic is the pressure in the cluster?

We measure pressure in our system by considering the contributions from particle-particle collisions and particle-wall collisions. As part of the pressure formula (given in Methods), the packing fraction of the system has to be known. While collisions can be analyzed locally (even for individual particles), the calculation of pressure as a function of the position requires the measurement of a local packing fraction. Unfortunately, local packing fraction is not well-defined for small regions of the system. For this reason we did not attempt to calculate pressure as a function of position. Furthermore, computing the pressure tensor is not straightforward for hard particles [S. Dussi, M. Dijkstra, Nature Comm. 7, 11175 (2016)]. Besides, in event-driven molecular dynamics simulations, the rate of pressure equilibration is determined by the speed of sound – typically such equilibration is rapid [D. Frenkel, Euro. Phys. J. Plus 128, 10 (2013)]. We therefore expect the pressure to be homogeneous and isotropic even though we did not explicitly demonstrate that this is the case. Note that for the arguments in our manuscript, homogeneity and isotropy of pressure is not required.

Reviewer 3

The authors use microfluidic techniques to bind nanoparticles into emulsion droplets, which are subsequently evaporated to yield clusters of nanoparticles. Using computer simulation, they find that such particles with excluded volume interactions exhibit “magic numbers” in that certain cluster geometries are favoured from the point of view of the free energy. The work is potentially interesting, though there is some considerable way to go before publication can be considered.

We thank the reviewer for finding our work potentially interesting and appreciate the constructive feedback. Below are our responses to the reviewer’s comments.

It appears to me upon reading the manuscript that there are really two sides to this story. On the one hand are experiments. But apart from demonstrating the existence of Mackay and anti-Mackay icosahedra, I see little that is really new. Mainly this is because ref 17 already demonstrates the experimental technique. While Ref 17 reports icosahedral clusters, and seems to me truly groundbreaking (enough to get a mention on the “Big Bang theory” TV show), it does not discuss magic numbers. So the main development of this paper is magic number clusters. But those are only shown with simulation. Again, ref 17 used simulation to determine the minimum free energy clusters.

As the reviewer correctly identified, our story has two sides. The de Nijs paper (now, Ref. 22) already showed that icosahedral symmetry arises in confinement due to entropy. However, while the general occurrence of icosahedral symmetry is important and of fundamental interest, it does not provide any insights into structural details. Our manuscript expands upon the status quo along two main directions:

- *Firstly, in our manuscript we provide a complete geometric model that accurately describes the crystal structure for colloidal clusters in a number range from a few hundred to about ten thousand particles. We support this geometric model by characterization of the surface structure (which enables a precise determination of the type of structure) and electron tomography (extended in the new Figure 3). As this, we believe that our contribution significantly extends the understanding of confined colloidal clusters in experiment.*
- *Secondly, we reveal for the first time that these clusters follow a discrete series, which justifies the term “magic”. High-precision free energy calculations show free-energy wells for certain ranges of particle numbers.*

While we use simulations to systematically understand the system (including kinetic pathways and thermodynamic minimum energy structures), we support all findings by experiments. Figure 4 provides a detailed overview of observed magic number structures that clearly show a very high degree of order coinciding with the simulated structures. The electron tomography data further supports the structural similarities between experiments and simulations.

In short, the authors have used established methods to infer that Mackay and anti-Mackay icosahedra are magic number clusters. Given the extensive literature on atomic clusters (e.g. Baletto and Ferando Rev. Mod. Phys. 77 371 (2005)), which I think the authors should cite, this isn't really a big surprise. While indeed the authors have a confined system, rather than attractive atoms, the similarity in structure between the (admittedly smaller) colloidal clusters made in the pioneering work on such confined systems of Manoharan (ref 14) and atomic clusters already hints strongly that magic numbers should also be seen for nanoparticle/colloidal clusters. Indeed, such a result seems unavoidable from the famous WCA paper (J. Chem. Phys. 54, 5237 (1971)).

We thank the reviewer for the references, which we added to the main text body. It is indeed interesting to see the structural similarities in atomic clusters and our observations. We feel that the connections between atomic clusters and colloidal clusters were not made clear in our original submission and extended the introduction to discuss similarities and differences of the two systems:

“Deviations from a pure statistical distribution of cluster sizes are also observed in the mass spectrum of small noble gas and metal clusters (3, 4). Clusters with atom numbers that follow magic numbers show enhanced stability as a result of the maximized numbers of neighbors (5–7). This tendency provides a driving force to form cluster with closed, concentric shells, including the famous Mackay icosahedra (8). Atomic clusters therefore often possess complex geometries that differ from the typical face-centered cubic (FCC) bulk crystal structure (9, 10).

In the colloidal realm, individual colloidal particles need to be stabilized against aggregation to form ordered assembly structures (11). Attractive interactions are minimized and the phase behavior of colloids can often be modelled with non-interacting hard sphere models that crystallize as a result of entropy maximization upon a critical packing fraction (12, 13). The formation of clusters with a defined number of constituent particles therefore requires a geometric confinement of stabilized particles (14–22). Temperature as the control variable for atomic cluster formation is replaced by the packing fraction of the colloids within the confined volume (23–27). Whereas the structure of atomic clusters is commonly explained by potential energy minimization, especially at low temperature (28), confined self-assembly of nanoparticles and colloids is driven by a combination of principles. Phenomena that contribute to the structure of a colloidal cluster include attractive interactions of soft ligand shells (29) or the presence of depletants (30), capillary forces acting on particles during the drying process (15), and entropy maximization (22).”

We politely disagree with the assessment concerning the interpretation of our results in the following aspects:

- We did not establish that Mackay icosahedra are magic numbers. We introduced these clusters along with an overview of the field of atomic clusters already in the abstract and introduction and added more descriptions in the revised version.*
- We did discover that such magic clusters exist for a colloidal soft matter system, which we believe is a significant difference to an atomic system. The fact that these structures are similar to atomic clusters adds to the generality of the concept of magic clusters.*
- Our structures are in fact quite different from those in the Manoharan paper. This important paper shows the existence and preparation of very small and defined clusters (less than 10*

particles). In our manuscript, the number of particles within the clusters are between a few hundred and 10,000. In their case, the symmetry is dictated by packing (as capillary forces draw the structures together). In our case, we (and de Nijs et al.) show that entropy governs structure formation.

- We provide evidence that a complex kinetic pathway is required to form well-ordered clusters and combine kinetic investigations with thermodynamic characterization of the minimum-energy structures. A similar analysis has not been performed to the best of our knowledge in soft matter nor atomic clusters.
- The WCA paper does not consider finite systems. While demonstrating that structures can result in a liquid, it is not clear to us how it predicts the existence of magic numbers with defined shells in a colloidal soft matter system.

The authors use an excessive amount of hyperbole in their manuscript. One example (there are more!) is that the statement “Our findings demonstrate that the concept of magic numbers is not limited by the types of interactions and much more ubiquitous than previously believed” badly needs to be toned down. One more interaction potential (hard sphere) on top of the very many existing, for example in the Cambridge Cluster database, can hardly be grounds for a “much” higher level of ubiquity. Magic numbers in the sense of shells which the authors seem to use aren’t exactly ubiquitous anyway, clusters of carbon being an obvious example! The authors could help their case, in terms of the generality of their work by citing C60 clusters, which also exhibit the kind of magic numbers they find with the nanoparticles - see, again, Baleto and Ferando Rev. Mod. Phys. 77 371 (2005).

We thank the reviewer for the critical assessment. We have taken care to remove any exaggerated statements throughout the text, for example in the following passages and included a clearer discussion on other systems (C60 clusters (ref. 56), micelles (Ref 55), ionic liquid supramolecules (Ref 58) and inorganic molecular assemblies (Ref 59).

(abstract) “Here we show that magic number clusters exist with superior stability in a confined soft matter system. Colloidal particles in an emulsion droplet spontaneously organize into a series of clusters with precisely defined shell structures. Crucially, free energy calculations demonstrate that colloidal clusters with magic numbers possess higher thermodynamic stability than those off magic numbers. A complex kinetic pathway is responsible for the high efficiency of this system in finding its minimum free energy configuration. Targeting similar magic number states is a strategy towards unique configurations in finite self-organizing systems across the scales.”

(introduction) “Small numbers of nanoparticles and colloids pack into defined clusters with a wide variety of symmetries (30–33). With increasing particle number, entropy maximization has been identified as the driving force favoring the formation of icosahedral symmetry of spherical colloidal clusters (22, 34). Importantly, the formation of such icosahedral clusters can be reproduced by simulations using hard spheres, demonstrating that interactions between individual particles are not necessary for the crystallization process (22), which is dominated by entropic effects. In such confined systems, the interface boundary acts as a source for heterogeneous nucleation (35) of crystalline patches (36) that subsequently expands towards the cluster center (22). If the number of particles within the confining element is further increased, the effect of confinement is continuously reduced until, from a system size of approximately 100,000 particles, bulk crystal structures with FCC symmetry are retained (22). Despite these general insights, the detailed structure of a large colloidal cluster formed from a discrete number of colloidal particles remains unknown. While the dominant occurrence of atomic clusters with magic numbers in mass spectra (3, 4) suggests self-adjustment in their sizes during formation by addition or removal of atoms to reach magic numbers

under appropriate circumstances, the number of particles in a colloidal cluster remains fixed in confinement, which allows experiments to explore clusters of arbitrary sizes. This raises the fundamental question whether some colloidal cluster sizes are thermodynamically preferred over others.”

(conclusion) “The existence of MCCs reported in this work has similarities to other clustering processes, such as micelles (55), fullerene assemblies (56), protein aggregates (57), supramolecules of ionic liquid (58), inorganic clusters (59), and natural framboidal pyrite (60, 61). Just like these natural self-organization processes, MCC formation is efficient in solving the high-dimensional optimization problem of finding a global free energy minimum. Our combined experimental and simulation data suggests that heterogeneous nucleation and subsequent restoration of mobility near the confinement interface, reminiscent of a lubrication layer, is essential for the high structure quality of MCCs. Similar design principles may be applied to hierarchical structures of bimetallic nanoparticles (48), molecular self-assembly (62), and plasmonic nanocluster formation (63). It remains to be investigated if MCCs can undertake structural transition towards other symmetries as observed in atomic clusters (64), and if magic number effect exists in other finite soft matter systems when the shape (33, 65), length scale (66), or rigidity of confinement (67) is varied.”

In summary, the authors need to tone down their manuscript and be very much more upfront about what they have actually achieved. That they have used simulation to predict magic number clusters in colloidal/nanoparticle systems is interesting, and that they have observed such clusters experimentally is noteworthy. They do not seem to show any statistics in their experiments: I realise that is hard with this technique, but something along the lines of the breakthrough work of Meng et al. (Science 327 560 (2010)) would help their case considerably.

We thank the reviewer for his/her positive and constructive advice and revised the manuscript accordingly (see comments above). Obtaining experimental statistics on the magic number effect is technically demanding, as the reviewer points out. The work of Meng (Ref. 30) involves up to 10 micron-sized particles, where observation of individual configurations via a microscope is possible. In the case of large colloidal clusters, there are three main difficulties to use the same approach:

- *The first difficulty comes from placing the exact number of colloids in a large emulsion droplet.*
- *The second difficulty comes from small particle size (about 250 nm), which excludes the use of an optical microscope for direct observation. The small size is necessary to access the time scales required to equilibrate into magic clusters and can therefore not be scaled up into the micron scale.*
- *The third difficulty is that the large size of the cluster prevents determining the interior structure from observation of the surface (as in the Meng paper). Statistics via electron tomography is experimentally not feasible because of the high time requirements for sample preparation, imaging and reconstruction.*

With our best effort, we examined over 1500+ colloidal clusters in the SEM to illustrate the cluster formation at fixed system size, and observed over tens of thousands of clusters at different sizes to identify over 25 magic number cluster structures that show excellent agreement with model prediction, 3D reconstruction, and simulation.

Reviewers' comments:

Reviewer #1 (Remarks to the Author):

The authors have satisfactorily answered to the comments. I recommend publication in Nature Communications

Reviewer #2 (Remarks to the Author):

The authors have improved the readability of the manuscript considerably and have addressed all the comments/remarks of all reviewers. I hereby recommend publication as is.

Reviewer #3 (Remarks to the Author):

The authors have made some efforts to address the comments made in the first round of reviewing. However, in the process their introduction has become frankly confusing and will need considerable work, more significantly it has emerged that whether they have really made a groundbreaking development seems rather questionable: their experiments reproduce those of previous work (ref 22) - with more statistics - but the results are expected. And I have concerns about their simulation analysis, see below.

Why do the authors think that their event driven MD simulations are representative of Brownian colloidal particles with hydrodynamic interactions? In the bulk, hydrodynamics have been shown to have a profound effect on crystallization (see e.g. Radu and Schilling EuroPhys. Lett., 2014, 105, 26001). Given the importance of hydrodynamics, do the authors seriously think that their _Newtonian_ dynamics are appropriate to this system in the context of elucidating a kinetic mechanism for structure formation? The simulations start in the same place as the experiments, and must end in the same place if both reach equilibrium. But what, if anything can Newtonian dynamics, relevant to atoms, tell us about kinetic pathways in colloids?

The introduction as it stands is thoroughly unclear. It appears that the authors have made a rushed attempt to address comments in the first round of reviewing, without really thinking carefully about the structure of the introduction. Atoms and colloids are mixed up, and the nature of the effective interactions between colloids is not correctly explained. For example the phrase " Attractive interactions are minimized " may be relevant for experimental approximation so hard spheres, but what of depletion systems - such as those of ref 30? The sentence at the bottom of the same paragraph beginning "Phenomena that..." then contradicts the statement above!

Furthermore the authors state that their structures are "quite different" to those of the Manoharan 2010 Science paper. Are they really? The structures are larger, but the method used by Manoharan in his 2003 Science paper (ref 15) is actually essentially the same as that used here: remarkably the

structures are more or less identical to those in the 2010 paper - despite the difference in the interactions. The structures the authors consider are larger, and as far as I know, there are no calculations for "sticky spheres" of the kind of size the authors are considering, so I don't really understand their statement.

As for the ground-breaking nature of the research: the experiments, we are agreed have been previously demonstrated (ref 22). Given that colloids are well-known to interact in analogous ways to atoms and to obey statistical mechanics, magic number clusters must be found. The reason for the reference to the WCA paper, which the authors seem to have missed, is to point out that WCA-type systems form structures almost identical to those in attractive systems, if some field is employed which may bring the system to the same density. Here the field is confinement. Therefore, knowledge of the behavior of Lennard-Jones clusters, now some 30 years old, combined with the WCA work, means that the behavior that the authors find is expected. So in short, the authors have taken an existing experimental result, considered statistical more data, and found that there are magic numbers in the system, as one would predict from WCA+previous work on attractive clusters in atomic systems. Rather than a discovery, the authors have obtained an experimental verification of an expected phenomenon, and they need to be upfront about this.

Reviewer 1

The authors have satisfactorily answered to the comments. I recommend publication in Nature Communications

We thank the reviewer for the positive evaluation and recommendation for publication in Nature Communications.

Reviewer 2

The authors have improved the readability of the manuscript considerably and have addressed all the comments/remarks of all reviewers. I hereby recommend publication as is.

We thank the reviewer for the positive evaluation and recommendation for publication in Nature Communications.

Reviewer 3

The authors have made some efforts to address the comments made in the first round of reviewing. However, in the process their introduction has become frankly confusing and will need considerable work, more significantly it has emerged that whether they have really made a groundbreaking development seems rather questionable: their experiments reproduce those of previous work (ref 22) - with more statistics - but the results are expected. And I have concerns about their simulation analysis, see below.

We thank the reviewer for acknowledging our efforts and the critical evaluation. We have revised the manuscript. Below are our point-by-point responses to the reviewer's comments.

Why do the authors think that their event driven MD simulations are representative of Brownian colloidal particles with hydrodynamic interactions? In the bulk, hydrodynamics have been shown to have a profound effect on crystallization (see e.g. Radu and Schilling EuroPhys. Lett., 2014, 105, 26001). Given the importance of hydrodynamics, do the authors seriously think that their Newtonian dynamics are appropriate to this system in the context of elucidating a kinetic mechanism for structure formation? The simulations start in the same place as the experiments, and must end in the same place if both reach equilibrium. But what, if anything can Newtonian dynamics, relevant to atoms, tell us about kinetic pathways in colloids?

We thank the reviewer for pointing out the importance of hydrodynamic interactions. We are aware of the paper by Radu and Schilling and similar works discussing the role of hydrodynamics for colloidal structure formation. Indeed, hydrodynamics influences crystallization kinetics although the final equilibrium structure is usually much less affected. This is the case even for the highly non-equilibrium situation of colloidal gelation (arxiv:1808.01722). Therefore we believe that for our equilibrium results (Fig. 4, Fig. 5h) hydrodynamics is not crucial. Layer formation (Fig. 5g) is also well-known from simulations involving hydrodynamic interactions (Roehm et al. Soft Matter

10, 5503, 2014). Research suggests that hydrodynamics can both slow down and speed up crystallization. As we do not make quantitative predictions about crystallization speed, we believe that our computational approach, which is employed also in other recent work (e.g. the discovery of icosahedral colloidal clusters, *Nat. Mater.* **14**, 56 (2015)), is just one more simplification (among several others) that we need to introduce to make our simulations feasible given our available computational resources and that is justifiable for the points made in our manuscript. The following sentence and three new citations (52-54) were added to the manuscript to clarify these issues:

“[Colloids are modeled using event-driven molecular dynamics.] This computational approach follows recent work (25) and ignores hydrodynamic interactions, which affect crystallization speed (52, 53) but are expected to have a weak influence on equilibrium cluster structure (54).”

52. M. Radu, T. Schilling, Solvent hydrodynamics speed up crystal nucleation in suspensions of hard spheres. *Europhys. Lett.* **105**, 26001 (2014).

53. D. Roehm, S. Kesselheim, A. Arnold, Hydrodynamic interactions slow down crystallization of soft colloids. *Soft Matter.* **10**, 5503–5509 (2014).

54. J. de Graaf, W. C. K. Poon, M. J. Haughey, M. Hermes, Hydrodynamics strongly affect the dynamics of colloidal gelation but not gel structure (2018) (available at <http://arxiv.org/abs/1808.01722>).

The introduction as it stands is thoroughly unclear. It appears that the authors have made a rushed attempt to address comments in the first round of reviewing, without really thinking carefully about the structure of the introduction. Atoms and colloids are mixed up, and the nature of the effective interactions between colloids is not correctly explained. For example the phrase “ Attractive interactions are minimized ” may be relevant for experimental approximation so hard spheres, but what of depletion systems - such as those of ref 30? The sentence at the bottom of the same paragraph beginning “Phenomena that...” then contradicts the statement above!

We have taken great care to not mix the discussion of clusters from atoms and clusters from colloids. The first paragraph of the main text exclusively deals with atoms, while the following paragraph deals with colloids only. Indeed, the order of the sentences in the second paragraph has been confusing. We therefore restructured the second paragraph according to the reviewer’s suggestion to make it clearer:

“Whereas the appearance and structure of atomic clusters is commonly explained by potential energy minimization, especially at low temperature (11), the formation of colloidal clusters is typically governed by other factors. Colloids are initially stabilized against aggregation, which prevents them from self-assembling in solution (12,20). However, colloids can be forced into clusters by geometric confinement (13–22). Phenomena that contribute to the development of colloidal clusters during such a confined self-assembly process include the interaction of soft ligand shells (23) or the presence of depletants (24), capillary forces acting on particles during the drying process (14), and entropy maximization (21). The latter case is comparably easy to control in experiment because it only requires weakly interacting colloids, which are described well via the simple hard sphere model (25, 26). Temperature as the control variable for atomic cluster formation is then replaced by the packing fraction of the hard spheres within the confined volume (27–31).”

Furthermore the authors state that their structures are “quite different” to those of the Manoharan 2010 Science paper. Are they really? The structures are larger, but the method used by Manoharan in his 2003 Science paper (ref 15) is actually essentially the same as that used here: remarkably the structures are

more or less identical to those in the 2010 paper - despite the difference in the interactions. The structures to authors consider are larger, and as far as I know, there are no calculates for “sticky spheres” is the kind of size the authors are considering, so I don’t really understand their statement.

To clarify the differences among our and Manoharan’s work, we make comparisons in the following figure and table.

Figure 1. *a*, The clustering process of Manoharan 2003. Particles adhered at the interface are dragged together as the droplet volume shrinks. Many cluster symmetries are found. **b**, Brenner (Ref 51, PRL, 2004) showed that cluster structures were determined by wetting and particle packing at the droplet interface. **c**, In Manoharan 2010, colloidal cluster form with short-range attraction induced by depletion without confinement. Their key finding is that rotational entropy disfavors high-symmetry clusters (octahedron occurs less frequently than poly-tetrahedron for a cluster of 6 particles). Because the number of possible configuration increases exponentially with the number of particles, the authors did not study clusters made from 12 spheres, the smallest number to form an icosahedral cluster. Still, they concluded that icosahedral symmetry is very unlikely to appear. **d**, Exemplary cluster in our work. Our clusters have icosahedral symmetry and consist of 100 to 8000 particles.

	Number of particles	Clustering process	Relevant conclusion
Manoharan 2003	< 12	Particle packing at the interface	Minimization of second moment of mass
Brenner 2004	< 15	Particle packing at the interface	Wettability and packing at interface determines structures
Manoharan 2010	< 11	Depletion attraction	High symmetry structure is suppressed by rotational entropy
This work 2018	100 – 8000	Entropy maximization under spherical confinement	I_h symmetry is favoured; clusters of some sizes are more stable than clusters of other sizes

The definition of magic numbers is only sensible when comparisons among clusters of different sizes is made. Manoharan 2003 aimed at finding the minimization principle of small numbers of spheres trapped at the droplet interface, essentially a two-dimensional packing problem. Manoharan 2010 compared distinct configurations of equal-sized colloidal clusters. Apart from the difference in system size as well as the dominating cluster symmetry and clustering mechanism (all of which differ from what we find in our system), a major difference of our work to the ones cited by the referee is that we compare clusters across sizes, whereas previous works focused on comparing clusters of fixed size with another.

As for the ground-breaking nature of the research: the experiments, we are agreed have been previously demonstrated (ref 22). Given that colloids are well-known to interact in analogous ways to atoms and to obey statistical mechanics, magic number clusters must be found. The reason for the reference to the WCA paper, which the authors seem to have missed, is to point out that WCA-type systems form structures almost identical to those in attractive systems, if some field is employed which may bring the system to the same density. Here the field is confinement. Therefore, knowledge of the behavior of Lennard-Jones clusters, now some 30 years old, combined with the WCA work, means that the behavior that the authors find is expected. So in short, the authors have taken an existing experimental result, considered statistical more data, and found that there are magic numbers in the system, as one would predict from WCA+previous work on attractive clusters in atomic systems. Rather than a discovery, the authors have obtained an experimental verification of an expected phenomenon, and they need to be upfront about this.

Concerning the novelty of this work, we observe a family of icosahedral clusters spanning a large range of particle numbers in the experiments (a few observed in atomic clusters, many not) and propose a new geometric model to accurately describe all observed cluster structures. Our simulations reproduce the experimental observations, and demonstrate previously unknown details of the kinetic pathway for cluster formation in confinement. We apply non-trivial free energy calculation of a kind and precision never performed before in any comparable system or situation to illustrate the magic number phenomena in colloidal clusters. None of these findings are included in Ref. 22 (current manuscript, Ref. 21). We do not understand how the WCA paper (J. Chem. Phys. 54, 5237 (1971)), together with Lennard-Jones clusters, can deduce the magic number effect in colloids. To the best of our knowledge, since the discovery of magic number in nuclei in 1949 and atomic clusters in 1981, which all have been properly acknowledged in our manuscript, we are not aware of any predictions or reports of the magic number phenomenon where the building block is in the colloidal length scale.

REVIEWERS' COMMENTS:

Reviewer #1 (Remarks to the Author):

I am asked to judge about the reply of the authors to the points raised by Referee 3. Referee 3 raises four points, which have been taken into account by the authors in their revision.

1) The first point is about neglecting hydrodynamic interactions in the simulations. The author correctly reply noting that hydrodynamic interactions are not important for determining the equilibrium cluster structures, whereas they might be important for modelling formation kinetics. The reply is convincing.

2) The second point is about the introduction, which is unclear according to Referee 3. The authors have thus rewritten a paragraph and now the introduction is definitely improved. I find it clear and well written.

3) The third point is about novelty, in particular about whether the structures are "quite different" from those found previously in the literature, specifically by Manoharan et al.. Referee 3 states that the structures are "more or less identical to those in the 2010 paper" by Manoharan et al. (Ref. (24)). I have personally checked both the 2010 and the 2003 paper by Manoharan et al., and I cannot see how the structures shown in these papers can be qualified as "more or less identical" to those of the present paper. The structures of the present paper contain several thousand colloids, whereas the structures by Manoharan et al. contain ~10 colloids. Moreover, the structures in Manoharan et al (2010) are quite different, since, as written in Manoharan et al. (2010) itself "Structures with fivefold symmetry, such as the pentagonal dipyramid and icosahedron, are highly unfavorable in our system". In the present paper, most structures are of the icosahedral family, with some non-trivial variants such as the anti-Mackay icosahedron and the pentakis dodecahedron. The authors of the present paper are fully convincing in their reply, which is very well supported by the images from the different papers involved in this discussion, as reported in the figure and in the table of their reply.

4) The last point concerns the fact that the overall findings of this paper may be expected, on the basis of the following statement of Referee 3:

"Given that colloids are well-known to interact in analogous ways to atoms and to obey statistical mechanics, magic number clusters must be found." This is indeed not at all warranted, especially when the number of particles in the clusters (i.e. the cluster size) becomes large. Icosahedral structures are inherently strained, with strain increasing with size. This may lead to a symmetry breaking of the structures, in which the magic character might be lost (see J. Phys. Condensed Matter 27, 013003 (2015)). Moreover: even if high symmetry is retained, which magic numbers? Those of Mackay icosahedra, anti-Mackay icosahedra and pentakis dodecahedra are different. Finally, I am not aware of any demonstration of the production of such non-trivial magic-number clusters at the same level of precision for such large sizes, with several thousand particles per cluster, even in atomic systems. Therefore I judge that the results of this paper are indeed groundbreaking and, to a large extent, quite surprising.

In summary, I judge that the authors' reply is fully convincing and I recommend publication in Nature Communications without further delay.

Reviewer #2 (Remarks to the Author):

I already recommend publication in Nature Communications, but read anyway the critiques of Reviewer 3. With the help of the comments of Reviewer 3, the introduction has been improved in the revised manuscript. I agree with the authors that hydrodynamic interactions may play a role in the kinetics of the crystallization, but will likely not affect the resulting equilibrium structures. This has indeed been shown by the good comparison of the final structures as obtained from experiments and from simulations in this work, but also in Nat. Mater. 14, 56 (2015) and Nature Communications 8, 2228 (2018), and even in attractive systems Nano Letters 18, 3675-3681 (2018). I also agree with the authors that the clusters studied by Manoharan are very different, not only the size of the clusters but also the formation mechanism. I find it rather surprising that the icosahedral symmetry is preserved for clusters up to 8000 spheres and that magic cluster sizes are favoured for these high particle numbers. I also don't see the relation of this work with the WCA reference, which do not form finite clusters because the interactions are purely repulsive and the Lennard-Jones clusters, which are stabilized by attractions. The magic numbers arise here due to confinement. In conclusion, the work is sound, the authors have performed a thorough study on the magic number clusters that arise as minimum free-energy structure, and extend considerably previous studies in this area.

Reviewer #3 (Remarks to the Author):

The authors make the statement "Indeed, hydrodynamics influences crystallization kinetics although the final equilibrium structure is usually much less affected". This is obvious and irrelevant. For sure their MD simulations represent the equilibrium situation. But that isn't what they are claiming - they state that their simulations can somehow reveal the mechanism by which the system moves towards equilibrium. This, by definition is a non-equilibrium process. And it is this non-equilibrium process that the authors are claiming is unaffected by hydrodynamic interactions. Yet they have precisely zero evidence in support of this.

What they need to do, as stated in the previous round of reviewing, is to be honest and upfront and simply state the limitations of their simulation technique. In no sense is it guaranteed to reproduce the experimental system's approach to equilibrium. That is to say, there is no evidence that the system should follow the same path through the energy landscape - due to the effect of hydrodynamics, not to mention the fact that even without the long-ranged many-body coupling induced by the Stokesian hydrodynamic interactions, even Langevin dynamics (i.e. only considering the solvent as random noise and making the very significant approximation of neglecting the many-body interactions) is in no sense guaranteed to reproduce the path through the energy landscape that either the real system would take, nor indeed the path that purely Newtonian dynamics would take.

This effect has been studied seriously, in addition to the work of Radu and Schilling, Tanaka has a series of papers which properly consider the effect of hydrodynamics on the non-equilibrium kinetics of colloids, e.g. PRL 104, 245702 (2010). Figure 4 of that paper indeed makes it very clear that hydrodynamics significantly alter the structures that a colloidal system passes through on its approach to forming an equilibrium cluster. Why on earth do the authors presume, with no justification, that their Newtonian dynamics can reveal the mechanism - i.e. route through the energy landscape - when such plainly contradictory observations have been made in the literature?

Note also that Ref 54 is unpublished work and exhibits a number of shortcomings, not least that it ignores the higher-order structure such as the icosahedra symmetry that the authors of the present manuscript consider, and that is so important in its interplay with hydrodynamics that was illustrated by Tanaka in his 2010 PRL.

The authors claim that they have “restructured the second paragraph according to the reviewer’s suggestion to make it clearer”. I wish they had! We hear that colloidal clusters usually self-assemble through confinement. Remarkably, refs 13, 15 and 16 don’t actually mention the word cluster in the context of colloids in the manuscript! (ref 15 mentions the atomic clusters from which the nano particles are grown - hardly relevant in this context). These papers are on crystallisation of nanoparticles.

Can it be that the authors are claiming that colloidal crystallisation through confinement, necessary in the case of hard spheres and charged colloids, is somehow pertinent to colloidal clusters? Perhaps they should also cite the original 1986 Pusey and van Meegen Nature paper on hard sphere crystallisation! Anyway, it would help the reader of this article if the authors could make some effort not to confuse work on colloidal crystallisation with the formation of clusters.

Mysteriously, the authors have somehow missed a considerable number of papers which, unlike those they have cited, pertain to colloidal clusters:

Bonn et al PRL 103, 156101 (2009)
de Hoog et al. Phys. Rev. E. 64, 021407 (2001)
Dibble et al. Phys. Rev. E. 74, 041403 (2006)
Campbell et al. Phys. Rev. Lett., 2005, 94, 208301
Wilking et al PRL 96, 015501 (2006)
Stradner et al Nature 432, 492-495 (2004)

I could readily continue, but I don’t think it is up to me to carry out a literature search for the authors!

Even more remarkably, in every one of these papers, the clusters are formed via attractions, sometimes induced via depletion, but nevertheless, these are thermodynamically equivalent to the attractive interactions which drive clustering in atomic and molecular systems. In other words, while clustering can be induced through confinement, as shown by ref 14, in colloids a large swathe of the literature concerns clustering driven by attraction: to suggest, as the authors do, that “Whereas the appearance and structure of atomic clusters is commonly explained by potential energy minimization...the formation of colloidal clusters is typically governed by other factors” is factually wrong and seems to be an attempt to suggest that there is something different in principle in the clustering of colloids, which there isn’t necessarily,

Regarding their claims that their work is somehow fundamentally different to that of Manoharan and that there is some difference in the geometries formed: the earlier work largely considered smaller clusters of <12 particles. Icosahedra require 13 at least. So, naively, it isn’t hugely surprising that Manoharan didn’t observe many icosahedra. Remarkably Fig 3b in the 2003 Science paper shows a cluster which seems remarkably similar to an icosahedron.

Statements the authors make that the geometries of their system generates are somehow different to those of Manoharan have no basis as far as I can see. Moreover I don't see how their system is meaningfully different compared to the 2003 paper: both are colloidal particles in liquid droplets, which are brought together by the change in size of the droplet. Please can the authors explain what is different between these two systems, save for the number of particles concerned? That is to say the authors claim that "Brenner showed that cluster structures were determined by wetting and particle packing at the droplet interface" - this may be relevant for small clusters, but clearly cannot hold for clusters of the size the authors consider. So what is it, other than the size, that the authors think is fundamentally different about their system?

I remain bewildered by the authors' insistence that the structures formed are different with respect to the Manoharan 2003 and 2010 papers in the Fig. 1 of their most recent rebuttal letter. Curiously, the authors have chosen not to show all the structures from either paper. Could this be because they didn't want to discuss that many of the structures are actually the same?

Let us go through the structures of the 2003 and 2010 papers:

5 particles: both systems form triangular bipyramids

6 particles: 2003 paper shows octahedra, 2010 paper has a competition between octahedra and polytetrahedra

7 particles. both systems form pentagonal bipyramids

8 particles. 2003 paper shows D_{2d} symmetry, which is also found in the system of the 2010 paper.

Moreover, all of these structures are found for attractive atomic systems: may I refer the authors to Doye et al J. Chem. Phys. 103, 4234 (1995).

In short, the authors have made a truly awful job of addressing the previous points. Can they make some kind of effort to actually get the introduction correct? That is to say:

(1) colloidal clustering is driven by attractions and confinement, see references above. The former is similar to atomic systems, and leads to similar structures, (the relative range of the interaction notwithstanding). See Doye et al J. Chem. Phys. 103, 4234 (1995), and Manoharan's work.

(2) hydrodynamics have a profound effect on the structure of non-equilibrium colloidal systems (hardly surprising if, as the authors acknowledge, the rate of certain processes is drastically altered). See work of Tanaka for example. This means that it is not possible to concretely infer a mechanism using Newtonian dynamics, as the authors seem to think is somehow possible. A further important comment here is that HI are more important at lower colloidal volume fractions, i.e. when the authors begin their compression of the droplets. This isn't the case with hard sphere crystallisation, in other words the work of Tanaka's 2010 PRL likely is more pertinent their system.

(3) as stated in the previous round of reviewing, please can the authors either explain properly rather than why their system is truly different to that of the Manoharan 2003, other than the number of particles? (i.e. making a decent like-for like comparison, without choosing for example different numbers of particles and not mentioning when the Manoharan system exhibits the same symmetries that the authors find?). Or if indeed it isn't very different, as appears to be the case, can they just say so?

Reviewer 3

The authors make the statement “Indeed, hydrodynamics influences crystallization kinetics although the final equilibrium structure is usually much less affected“. This is obvious and irrelevant. For sure their MD simulations represent the equilibrium situation. But that isn't what they are claiming - they state that their simulations can somehow reveal the mechanism by which the system moves towards equilibrium. This, by definition is a non-equilibrium process. And it is this non-equilibrium process that the authors are claiming is unaffected by hydrodynamic interactions. Yet they have precisely zero evidence in support of this.

What they need to do, as stated in the previous round of reviewing, is to be honest and upfront and simply state the limitations of their simulation technique. In no sense is it guaranteed to reproduce the experimental system's approach to equilibrium. That is to say, there is no evidence that the system should follow the same path through the energy landscape - due to the effect of hydrodynamics, not to mention the fact that even without the long-ranged many-body coupling induced by the Stokesian hydrodynamic interactions, even Langevin dynamics (i.e. only considering the solvent as random noise and making the very significant approximation of neglecting the many-body interactions) is in no sense guaranteed to reproduce the path through the energy landscape that either the real system would take, nor indeed the path that purely Newtonian dynamics would take.

No simulation is possible without suitable approximations. Here, approximations are the assumptions of (i) hard spheres, (ii) hard confinement, (iii) monodispersity, (iv) no hydrodynamics. We obtain our conclusions under these assumptions. All details of the model are clearly stated in the main text and method section. Future work may repeat our study using more sophisticated models.

This effect has been studied seriously, in addition to the work of Radu and Schilling, Tanaka has a series of papers which properly consider the effect of hydrodynamics on the non-equilibrium kinetics of colloids, e.g. PRL 104, 245702 (2010). Figure 4 of that paper indeed makes it very clear that hydrodynamics significantly alter the structures that a colloidal system passes through on its approach to forming an equilibrium cluster. Why on earth do the authors presume, with no justification, that their Newtonian dynamics can reveal the mechanism - i.e. route through the energy landscape - when such plainly contradictory observations have been made in the literature?

The key different to the work by Tanaka is that our structure formation occurs near equilibrium very slowly over the course of hours while gelation is far away from equilibrium. Hydrodynamics will be more important the further away a process is from equilibrium. We now include the paper by Tanaka as a new reference (Ref. 57) and write:

“This computational approach follows recent work²⁸ and ignores hydrodynamic interactions, which affect crystallization speed^{55,56} and colloidal aggregation far from equilibrium^{57,58} but are expected to have a weak influence on the equilibrium cluster structure and near-equilibrium structure formation.”

Note also that Ref 54 is unpublished work and exhibits a number of shortcomings, not least that it ignores the higher-order structure such as the icosahedra symmetry that the authors of the present manuscript consider, and that is so important in its interplay with hydrodynamics that was illustrated by Tanaka in his 2010 PRL.

We cite Ref. 54 (now Ref. 58) clearly marked as a preprint. Nature Communications explicitly allows the citation of arXiv preprints. It appears to us that the precise role of hydrodynamics is subject to ongoing research and a comment from our side is beyond to scope of the present work.

The authors claim that they have “restructured the second paragraph according to the reviewer’s suggestion to make it clearer”. I wish they had! We hear that colloidal clusters usually self-assemble through confinement. Remarkably, refs 13, 15 and 16 don’t actually mention the word cluster in the context of colloids in the manuscript! (ref 15 mentions the atomic clusters from which the nano particles are grown - hardly relevant in this context). These papers are on crystallization of nanoparticles.

Can it be that the authors are claiming that colloidal crystallization through confinement, necessary in the case of hard spheres and charged colloids, is somehow pertinent to colloidal clusters? Perhaps they should also cite the original 1986 Pusey and van Megen Nature paper on hard sphere crystallization! Anyway, it would help the reader of this article if the authors could make some effort not to confuse work on colloidal crystallization with the formation of clusters.

Mysteriously, the authors have somehow missed a considerable number of papers which, unlike those they have cited, pertain to colloidal clusters:

Bonn et al PRL 103, 156101 (2009)

de Hoog et al. Phys. Rev. E. 64, 021407 (2001)

Dibble et al. Phys. Rev. E. 74, 041403 (2006)

Campbell et al. Phys. Rev. Lett., 2005, 94, 208301

Wilking et al PRL 96, 015501 (2006)

Stradner et al Nature 432, 492-495 (2004)

I could readily continue, but I don’t think it is up to me to carry out a literature search for the authors!

Even more remarkably, in every one of these papers, the clusters are formed via attractions, sometimes induced via depletion, but nevertheless, these are thermodynamically equivalent to the attractive interactions which drive clustering in atomic and molecular systems. In other words, while clustering can be induced through confinement, as shown by ref 14, in colloids a large swathe of the literature concerns clustering driven by attraction: to suggest, as the authors do, that “Whereas the appearance and structure of atomic clusters is commonly explained by potential energy minimization...the formation of colloidal clusters is typically governed by other factors” is factually wrong and seems to be an attempt to suggest that there is something different in principle in the clustering of colloids, which there isn’t necessarily,

The work by Pusey and van Megen (1986) had already been cited in the context of the hard sphere model for colloids. We made an effort to be more clear in the second paragraph adding the Refs. 14-17. There are two ways to aggregate colloids into clusters, attraction and confinement:

“Whereas the appearance and structure of atomic clusters is commonly explained by potential energy minimization, especially at low temperature¹¹, the formation of colloidal clusters is typically governed by several factors. Colloids are initially stabilized against aggregation, which prevents them from self-assembling in solution^{12,13}. However, colloids can aggregate into clusters by increasing short-range attraction¹⁴⁻¹⁷ or by applying geometric confinement¹⁸⁻²⁶.”

Regarding their claims that their work is somehow fundamentally different to that of Manoharan and that there is some difference in the geometries formed: the earlier work largely considered smaller clusters of <12 particles. Icosahedra require 13 at least. So, naively, it isn't hugely surprising that Manoharan didn't observe many icosahedra. Remarkably Fig 3b in the 2003 Science paper shows a cluster which seems remarkably similar to an icosahedron.

Statements the authors make that the geometries of their system generates are somehow different to those of Manoharan have no basis as far as I can see. Moreover I don't see how their system is meaningfully different compared to the 2003 paper: both are colloidal particles in liquid droplets, which are brought together by the change in size of the droplet. Please can the authors explain what is different between these two systems, save for the number of particles concerned? That is to say the authors claim that "Brenner showed that cluster structures were determined by wetting and particle packing at the droplet interface" - this may be relevant for small clusters, but clearly cannot hold for clusters of the size the authors consider. So what is it, other than the size, that the authors think is fundamentally different about their system?

I remain bewildered by the authors' insistence that the structures formed are different with respect to the Manoharan 2003 and 2010 papers in the Fig, 1 of their most recent rebuttal letter. Curiously, the authors have chosen not to show all the structures from either paper. Could this be because they didn't want to discuss that many of the structures are actually the same?

Let us go through the structures of the 2003 and 2010 papers:

5 particles: both systems form triangular bipyramids

6 particles: 2003 paper shows octahedra, 2010 paper has a competition between octahedra and polytetrahedral

7 particles. both systems form pentagonal bipyramids

8 particles. 2003 paper shows D_{2d} symmetry, which is also found in the system of the 2010 paper.

Moreover, all of these structures are found for attractive atomic systems: may I refer the authors to Doye et al J. Chem. Phys. 103, 4234 (1995).

In short, the authors have made a truly awful job of addressing the previous points. Can they make some kind of effort to actually get the introduction correct? That is to say:

(1) colloidal clustering is driven by attractions and confinement, see references above. The former is similar to atomic systems, and leads to similar structures, (the relative range of the interaction notwithstanding). See Doye et al J. Chem. Phys. 103, 4234 (1995), and Manoharan's work.

(2) hydrodynamics have a profound effect on the structure of non-equilibrium colloidal systems (hardly surprising if, as the authors acknowledge, the rate of certain processes is drastically altered). See work of Tanaka for example. This means that it is not possible to concretely infer a mechanism using Newtonian dynamics, as the authors seem to think is somehow possible. A further important comment here is that HI are more important at lower colloidal volume fractions, i.e. when the authors begin their compression of the droplets. This isn't the case with hard sphere crystallisation, in other words the work of Tanaka's 2010 PRL likely is more pertinent their system.

(3) as stated in the previous round of reviewing, please can the authors either explain properly rather than why their system is truly different to that of the Manoharan 2003, other than the number of particles? (i.e. making a decent like-for like comparison, without choosing for example different numbers of particles and not mentioning when the Manoharan system exhibits the same symmetries that the authors find?). Or if indeed it isn't very different, as appears to be the case, can they just say so?

A major difference is that we compare free energies among large colloidal clusters of different sizes (the magic number effect) while Manoharan's work focuses on certain system sizes without comparison among system sizes. Furthermore, Manoharan's 2003 work, as supported by Brenner (PRL, 2004), is a two-dimensional packing problem on the spherical surface while our work deals with three-dimensional packing and assembly inside the spherical confinement.

As the reviewer pointed out, in Manoharan's 2003 work many symmetries are found. But only one system with twelve spheres has icosahedral symmetry. In Manoharan's 2010 work, systems with more than 10 spheres were not studied due to increasing permutation. In Manoharan's own words: "[...] structures with fivefold symmetry, such as the pentagonal dipyramid and icosahedron, are highly unfavorable in our system. Therefore we do not expect icosahedra or other clusters with fivefold symmetry to be a structural motif in attractive hard-sphere gels or fluid cluster phases, where the attraction is short-ranged." In our system, all observed magic number clusters are without exception icosahedral.